# Leafhopper salivary vitellogenin mediates virus transmission to plant phloem

Yanfei Wang[1,2], Chengcong Lu[1,2], Shude Guo[1], Yuxin Guo[1], Taiyun Wei[1] & Qian Chen[1]

Salivary effectors of piercing-sucking insects can suppress plant defense to promote insect feeding, but it remains largely elusive how they facilitate plant virus transmission. Leafhopper *Nephotettix cincticeps* transmits important rice reovirus via virus-packaging exosomes released from salivary glands and then entering the rice phloem. Here, we report that intact salivary vitellogenin of *N. cincticeps* (NcVg) is associated with the GTPase Rab5 of *N. cincticeps* (NcRab5) for release from salivary glands. In virus-infected salivary glands, NcVg is upregulated and packaged into exosomes mediated by virus-induced NcRab5, subsequently entering the rice phloem. The released NcVg inherently suppresses $H_2O_2$ burst of rice plants by interacting with rice glutathione S-transferase F12, an enzyme catalyzing glutathione-dependent oxidation, thus facilitating leafhoppers feeding. When leafhoppers transmit virus, virus-upregulated NcVg thus promotes leafhoppers feeding and enhances viral transmission. Taken together, the findings provide evidence that viruses exploit insect exosomes to deliver virus-hijacked effectors for efficient transmission.

Approximately 80% of known plant viruses are transmitted by insect vectors, most of which are piercing-sucking Hemiptera insects such as leafhoppers[1–3]. The process of insect feeding involves plant defense and insect counter defense to plant resistance. When plants perceive these insect attackers, $Ca^{2+}$ signaling, biosynthesis of Jasmonate (JA), JA-Ile, salicylic acid (SA), and ethylene, reactive oxygen species (ROS) signaling are induced, and the production of volatile compounds is also triggered to regulate systemic defenses[4,5]. Insect attackers then suppress plant defenses by releasing effectors. Once plant viruses join the arms race between plants and piercing-sucking insects, viruses positively regulate insect adaptation to plant during viral acquisition by insect, or regulate insect behavior during viral transmission by insect[6]. Well-documented investigations focus on the mechanism that viral infection in plant regulates plant defense to improve insect adaptation, ultimately promoting the virus acquisition by insect[7–14]. In contrast, investigation on how viruses improve insect counter defense to plant is relatively limited. Recently, we reported that rice gall dwarf virus (RGDV) utilizes viral protein to reduce calcium-binding protein secretion in the salivary gland of leafhopper vector, causing substantial callose deposition, ultimately enhancing viral transmission[15]. It is believed that viruses can exploit more insect factors to enhance the insect counter defense to plant, which facilitates insect feeding and viral transmission.

Rice dwarf virus (RDV) is the first plant virus recorded to be transmitted by insect vectors in 1895. It causes severe rice dwarf disease in Asia and is mainly vectored by the leafhopper, *Nephotettix cincticeps*, in a persistent-propagative manner. *Nephotettix cincticeps* transmits RDV intermittently, potentially through a threshold-controlled viral release strategy[16]. When leafhoppers transmit RDV, the RDV exploits capsid protein P2 to interact with Rab5 of *N. cincticeps* (NcRab5), a small GTPase, in the exosomal pathway, allowing virus to hijack exosomes in salivary glands and traversing the apical plasmalemma into saliva-stored salivary cavities in a *N. cincticeps* Rab27a (NcRab27a)-dependent pathway. Combined with saliva, RDV-packaging exosomes reach rice phloem as leafhoppers feed, ultimately achieving viral transmission[17]. Exosomes are effective vectors

[1]Vector-borne Virus Research Center, State Key Laboratory of Ecological Pest Control for Fujian and Taiwan Crops, Fujian Agriculture and Forestry University, Fuzhou, Fujian, China. [2]These authors contributed equally: Yanfei Wang, Chengcong Lu. ✉e-mail: chenqian@fafu.edu.cn

for cell-to-cell delivery of biological factors and play important roles in cross-kingdom molecular exchange between hosts and pathogens for modulating host immunity and pathogen infection[18,19]. Therefore, exosomes are likely key transporters of virus-induced cross-kingdom factors from insect vectors to plant hosts, enabling plant adaption and facilitating viral transmission.

Vitellogenin (Vg), the precursor of yolk protein in insects, functions as the essential nutrient for oocyte development, and plays a role in immune defense, and viral transmission across insect generations[19–22]. Vg is also released to the plant host through insect saliva when piercing-sucking insects feed on plants[23]. The Vg of the small brown planthopper serves as an effector that weakens plant defense by interacting with rice immunity regulator OsWRKY71, while Vg in saliva or eggs of brown planthopper act as an elicitor of plant defenses by inducing the production of $H_2O_2$, JA, and JA-Ile[24,25]. However, it is unknown whether salivary Vg of other piercing-sucking insects functions during plant defense or viral transmission. The 220-kDa Vg of *N. cincticeps* (NcVg) contains 4 conserved domains, including 2 vitellogenin_N domains (NcVg1 and NcVg2), a domain of unknown function (NcVg3), and a von Willebrand factor type D domain (NcVg4)[26] (Fig. 1a). NcVg is synthesized by the fat body and transported to the hemolymph, where it is subsequently cleaved into 35- and 178-kDa subunits[26]. The 178-kDa subunit is taken up by oocytes in a receptor-dependent manner or by a novel bacterial symbiont-mediated manner that involves the obligate bacterial symbiont *Candidatus* Nasuia deltocephalinicola[26]. Proteomic data from Huang's and our labs have confirmed the presence of NcVg in saliva[23]. It is hypothesized that salivary NcVg may play a role in leafhopper feeding and may even function in RDV transmission.

In this study, we report that salivary NcVg associates with NcRab5 to release from leafhoppers and then functions as an effector suppressing $H_2O_2$ burst, thereby facilitating leafhopper feeding. RDV induces and hijacks the NcVg effector via NcRab5 in exosomal delivery pathway for release to the rice plant. The RDV-induced NcVg promotes the suppression of $H_2O_2$ burst in rice plants, facilitating viral transmission.

## Results

### NcVg releases to saliva cavities via the interaction with NcRab5

It is known that NcVg biosynthesized in the fat body is cleaved into 35-kDa and 178-kDa subunits in the hemolymph, and the 178-kDa subunit is taken up by oocytes[26]. We found the presence of intact NcVg (220 kDa) and the 178-kDa subunit in the whole body, and only the 178-kDa subunit in the ovary (Fig. 1b). The 220-kDa NcVg was also detected in the salivary glands and rice seedlings exposed to leafhoppers (Fig. 1b). These results indicate that intact NcVg is secreted from salivary glands and released to the rice host.

To investigate the manner of NcVg release into rice phloem, NcVg1, NcVg2, NcVg3 and NcVg4, which respectively covers 2 different vitellogenin N domains, a domain of unknown function, and a von Willebrand factor type D domain, were utilized as bait proteins to screen for their interactors from a cDNA library of *N. cincticeps* through the yeast two-hybrid system (Y2H). Of putative interactors of NcVg2, 39 sequences were annotated using the BLASTX analyses in the GenBank (Supplementary Fig. 1 and Supplementary Table 1). These putative interactors contained NcRab5, which regulates both intracellular and extracellular trafficking pathways and participates in RDV-packaging exosomes generation and delivery[17,27]. Further Y2H and glutathione S-transferase (GST) pull-down assays confirmed the interaction between NcVg2 and NcRab5 (Fig. 1c, d). The immuno-fluorescence microscopy of leafhoppers showed that the principal salivary gland was filled with apical plasmalemma-lined cavities. NcVg antigens were distributed as puncta and dispersed in the cytoplasm and salivary cavities (Fig. 1e). Colocalization of NcVg antigens with NcRab5 was observed in the cytoplasm and salivary cavities, some of which associated with apical plasmalemma (Fig. 1e). Western blot

assays on rice seedlings exposed to leafhoppers demonstrated the simultaneous presence of NcVg and NcRab5 (Fig. 1f). Immuno-fluorescence microscopy on sections of rice seedlings exposed to leafhoppers showed the colocalization of NcVg and NcRab5 in rice phloem (Fig. 1g and Supplementary Fig. 2a). Knockdown of *NcRab5* significantly reduced the NcRab5 and NcVg accumulation in the salivary glands and release to rice seedlings (Fig. 1h). These results suggest that NcVg associates with NcRab5 through interaction in the salivary glands and releases together into the cavities, ultimately reaching the phloem when leafhoppers feed on rice plants. In contrast, knockdown of *NcVg* had no significant effect on the transcription and expression of NcRab5 (Supplementary Fig. 2b, c). It was suggested the absence of feedback mechanism for association of NcVg and NcRab5.

Because Rab5 is also implicated in exosomes[28,29], whether NcVg was released via exosomes was investigated. The marker of exosomal membrane, NcRab27a, was used to determine the association of exosomes with NcVg. However, Y2H assays showed that NcVg2 failed to interact with NcRab27a (Fig. 1c). Immunofluorescence microscopy show most NcVg antigens did not colocalize with NcRab27a within the cytoplasm and salivary cavities (Supplementary Fig. 2d), suggesting that most NcVg antigens are released from salivary glands independent of the exosomal pathway in nonviruliferous leafhoppers.

### Salivary NcVg suppresses $H_2O_2$ production in rice as leafhoppers feed

To investigate whether NcVg is involved in regulation of rice defense, we first examined the insect-resistance response of rice plants. Exposure of rice plants to leafhoppers for 12 h significantly increased the contents of JA and the expression of JA-related genes, like *Allene oxide synthase* (*OsAOS*), but had no significant effect on the contents of SA or the expression of SA-, or ethylene-related genes, including *Enhanced disease susceptibility 1* (*OsEDS1)*, *Pathogenesis-related 1* (*OsPR1*), *Pathogenesis-related 5* (*OsPR5*) and *Ethylene-insensitive protein* (*OsEIN*) (Supplementary Fig. 2e, f). Leafhopper feeding also resulted in a significant increase in $H_2O_2$ content in rice seedlings, as well as the activity of $H_2O_2$ metabolism-related enzymes, such as peroxidase (POD) and catalase (CAT), and the accumulation of malondialdehyde (MDA), a metabolite of $H_2O_2$ (Supplementary Fig. 2g). The staining assays of 3,3′-diaminobenzidine (DAB), which indicates $H_2O_2$ accumulation, or 2′,7′-dichlorofluorescin diacetate (H2DCFDA), which is a fluorescent probe for $H_2O_2$, showed substantial $H_2O_2$ production in the feeding holes on rice leaves caused by exposure to leafhoppers (Supplementary Fig. 2h). These results indicate that exposure to leafhoppers for 12 h causes a ROS burst including $H_2O_2$ metabolism in rice plants.

Next, we examined the insect-resistance response of rice seedlings exposed to leafhoppers of which *NcVg* gene was knocked down. The results showed that in rice seedlings exposed to dsRNAs targeting *NcVg* (dsNcVg)-treated leafhoppers for 12 h, the release of NcVg was significantly decreased (Fig. 1i, j). However, the production of $H_2O_2$, as well as the activity of CAT and POD and the accumulation of MDA, were significantly increased compared to rice seedlings exposed to dsRNAs targeting GFP (dsGFP)-treated leafhoppers (Fig. 1k). DAB and H2DCFDA staining assays showed that knockdown of *NcVg* expression resulted in notably higher $H_2O_2$ production in the feeding holes and significantly higher number of feeding holes compared to dsGFP-treated leafhoppers (Fig. 1l, m). However, dsNcVg treatment had no significant effect on the contents of JA and SA, or the expression of JA-, SA-, or ethylene-related genes (Supplementary Fig. 3). These results indicate that NcVg is an effector that suppresses the $H_2O_2$ burst of rice plants.

The increased number of feeding holes caused by dsNcVg-treated leafhoppers suggested frequent probing and feeding. To analyze the feeding behavior of leafhoppers, electrical penetration graph (EPG) assays were performed. The waveforms produced by *N. cincticeps* feeding were categorized into Nc1-Nc4 (Supplementary Fig. 4), with Nc1 indicating non-probing, Nc2 indicating stylet penetration into

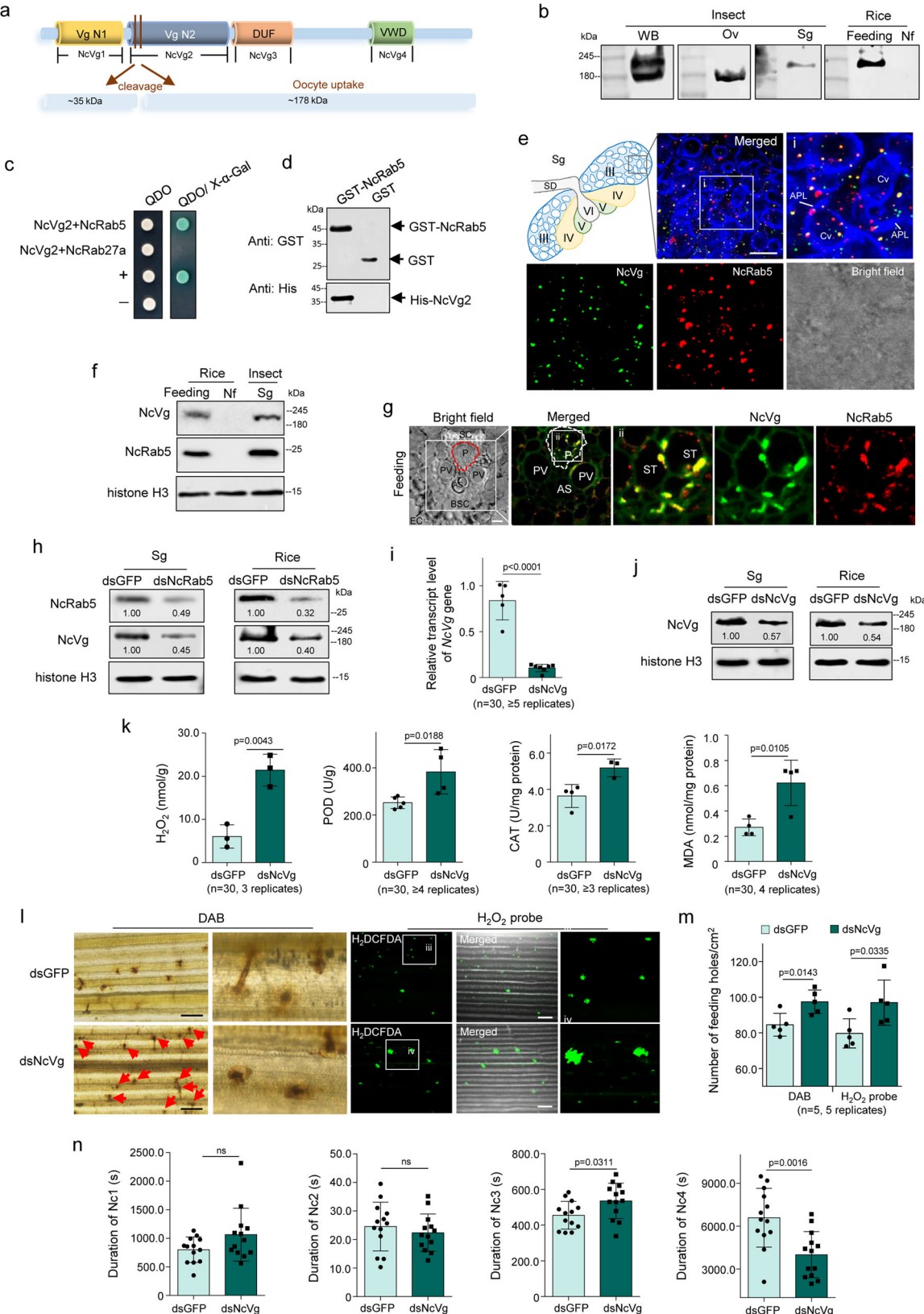

host, Nc3 signifying path and salivation, and Nc4 representing stylet presence in the phloem and xylem tissue[30–33]. Results showed that knockdown of *NcVg* expression significantly prolonged the duration of salivation (Nc3), and significantly shortened the duration of ingestion from phloem and xylem bundle tissues (Nc4), but had no effect on the duration of non-probing (Nc1) and stylet penetration (Nc2) (Fig. 1n).

Though the relative magnitude of the difference in Nc3 appears to be small, the statistically significant difference still suggested that dsNcVg-treated leafhoppers salivated longer to release more salivary proteins. Combined with data from Nc4, these results suggested that dsNcVg-treated leafhoppers encountered barriers when feeding, and suppression of $H_2O_2$ burst required more effectors in saliva. These

**Fig. 1 | Intact NcVg release from salivary gland to phloem for inhibiting $H_2O_2$ burst in rice plants. a** Domains and subunit composition of NcVg. **b** NcVg cleavage in organs as well as release to rice plant, as determined by western blot analysis. WB, whole body. Ov, ovary. Sg, salivary gland. Nf, non-feeding. **c** Y2H assays showing interaction of NcVg2 with NcRab5. +, positive control; −, negative control. QDO, SD/-Trp-Leu-His-Ade medium. **d** GST pull-down assays showing interaction of NcVg2 with NcRab5. **e** Distribution of NcVg and NcRab5 in cavities and cytoplasm of salivary gland, as determined by immunofluorescence microscopy. The schematic illustration of type III- to VI-cells in the salivary gland was shown. Panel **i** is the enlarged image of the boxed area. Panel showing green fluorescence (NcVg antigens), red fluorescence (NcRab5 antigens), or bright field is corresponding to panel i. APL, apical plasmalemma; Cv, cavity; SD, salivary duct. Bars, 10 μm. **f** Western blot assays of NcVg and NcRab5 in insect bodies and rice plants. **g** Distribution of NcVg and NcRab5 in rice phloem, as determined by immunofluorescence microscopy. Panels ii is enlarged image of the boxed area. AS, air space; BSC, Bundle sheath cell;

EC, Epidermis cell; P, phloem; PV, pitted vessel; SC, Sclerenchyma cell; ST, sieve tube. Bars, 10 μm. **h**, **i** and **j** Knockdown of *NcRab5* or *NcVg* expression reducing NcVg and NcRab5 accumulation in salivary glands and release to rice. Relative intensities of bands in western blot assays are shown. **k** Knockdown of *NcVg* expression inducing $H_2O_2$ burst and metabolism in rice plants, as determined by the content of $H_2O_2$ and MDA, as well as CAT and POD activity. **l** and **m** Knockdown of *NcVg* expression increasing the accumulation of $H_2O_2$ in feeding holes, as determined by DAB or H2DCFDA staining. The mean number of feeding holes per cm² of leaves are shown in **m**. Panels iii and iv are the enlarged images of the boxed areas. Bars, 200 μm. **n** Knockdown of *NcVg* expression increasing feeding difficulty of leafhoppers, as determined by EPG technique. Each dsNcVg- or dsGFP-treated leafhopper was continuously and electrically recorded during 3-hour feeding periods. Data in **b**, **d**, **e**, **f**, **g**, **h**, **i**, **j**, **k**, **l** and **m** represent at least 3 biological experiments. Data in **n** represent 13 valid biological replicates. Means ( ± SD) in **i**, **k**, **m,** and **n** are analyzed using two-tailed *t*-test. Ns, not significant.

results correspond to the higher production of $H_2O_2$ induced by exposure of rice plants to dsNcVg-treated leafhoppers. Therefore, it can be concluded that salivary NcVg is an effector that suppresses $H_2O_2$ production in rice plants, facilitating leafhoppers feeding.

## RDV hijacks NcVg to release via exosomal release pathway

A previous study has shown that RDV exploits the minor outer capsid protein P2 interacting with NcRab5 to hijack exosomes for its release from salivary glands to rice plants[17]. The interaction between NcVg and NcRab5 suggests that NcVg is likely induced by the P2-NcRab5 complex and is packaged in RDV-induced exosomes in the salivary glands of viruliferous leafhoppers. RT-qPCR and western blot assays showed that RDV infection significantly increased the transcript and protein expression levels of NcVg, NcRab5, and NcRab27a in salivary glands and increased the release of these proteins to rice plants (Fig. 2a, b). Immunofluorescence assays demonstrated the colocalization of NcVg with RDV, NcRab5, or NcRab27a in the cavities, cytoplasm, or associated with apical plasmalemma of salivary glands of viruliferous leafhoppers (Fig. 2c). Moreover, RDV infection in salivary glands increased the number of NcVg antigens colocalizing with NcRab5 or NcRab27a (Fig. 2c). Immunoelectron microscopy showed that NcVg antibody specifically colocalized with the virion in vesicles or membranes of vesicles within the cytoplasm or cavity of virion-containing salivary glands (Fig. 2d). The number of vesicles associated with NcVg antibody in cavities per section of one type III-cell of RDV-infected salivary glands was significantly higher than that of uninfected salivary glands (Fig. 2d). It was suggested that RDV infection likely caused salivary glands to secrete more NcVg to exosomes than to saliva. Combined with previous study revealing that RDV P2 protein interacts with NcRab5[17], these results imply that NcVg expression is indirectly induced by RDV-upregulated NcRab5. The P2-NcRab5-NcVg interaction, as well as the colocalization of RDV, NcVg and NcRab5, indicated that RDV-NcVg-NcRab5 form complexes. The complexes then hijack more exosomes for release from salivary glands to rice phloem for viral transmission. Immunofluorescence microscopy for sections of rice seedlings exposed to viruliferous leafhoppers illustrated the colocalization of NcVg with RDV, NcRab5, or NcRab27a in rice phloem (Fig. 2e). Knockdown of *NcRab27a* significantly decreased RDV, NcVg, and NcRab27a accumulation in salivary glands, as well as their release to rice seedlings (Fig. 2f). Treatment with GW4869, a specific exosome inhibitor, also blocked the accumulation of RDV, NcVg, and NcRab27a in salivary glands and the release to rice seedlings (Fig. 2g). These results indicate that NcVg is hijacked by RDV then released from salivary glands to rice phloem via exosomal pathway.

## Overexpression of NcVg in rice plants promotes leafhoppers to feed

Next, NcVg2-overexpressed (NcVg2-OE) transgenic plants were generated (Supplementary Figure 5a). Western blot assays showed that

NcVg2-OE plants expressed NcVg2 (35 kDa) in different extent (Supplementary Figure 5a and b). The lines #1 and #5 accumulating NcVg2 in higher levels were chosen for further analyses (Fig. 3a and Supplementary Figure 5b-d).

As shown in Fig. 3b, NcVg2-OE transgenic plants exposed or not exposed to leafhoppers demonstrated significantly lower production of $H_2O_2$, CAT and POD activity, and accumulation of MDA compared to WT plants. DAB and H2DCFDA staining assays showed that leafhoppers feeding caused a lower number of feeding holes in NcVg2-OE plants than in WT plants (Fig. 3c). These results suggest that overexpression of NcVg2 in rice plants likely decreases the frequency of probing and feeding. EPG assays also revealed that feeding on NcVg2-OE plants significantly shortened the duration of non-probing (Nc1) and salivation (Nc3), but prolonged the duration of ingestion from phloem and xylem bundle tissues (Nc4) (Fig. 3d). Thus, overexpression of NcVg in rice plants benefits leafhoppers feeding due to lower $H_2O_2$ production.

## RDV exploits NcVg effector to suppress $H_2O_2$ burst in rice plants

Next, the rice defense to viruliferous leafhopper feeding was investigated. The results showed that exposure to viruliferous leafhoppers for 12 h did not cause a significant change in JA or SA accumulation, or expression of JA-, SA- or ethylene-related gene in rice seedlings, compared to nonviruliferous leafhopper (Fig. 4a, b). In contrast, exposure to viruliferous leafhoppers for 12 h significantly increased $H_2O_2$ production, CAT and POD activity, and MDA accumulation in rice seedlings (Fig. 4c). DAB and H2DCFDA staining assays also revealed that exposure to viruliferous leafhoppers resulted in more $H_2O_2$ production in feeding holes and a higher number of feeding holes compared to nonviruliferous leafhoppers (Fig. 4d, e). EPG assays demonstrated that viruliferous leafhoppers feeding took significantly longer duration for salivation (Nc3), shorter duration for ingestion from phloem and xylem bundle tissues (Nc4) compared to nonviruliferous leafhoppers (Fig. 4f). These results indicate the increased difficulty of viruliferous leafhoppers feeding, and suggest that $H_2O_2$ burst is triggered by exposure to viruliferous leafhoppers.

The effect of knocking down *NcVg* expression on RDV release was then determined. RT-qPCR and western blot assays demonstrated that knockdown of *NcVg* expression in leafhoppers significantly decreased P8 accumulation in salivary glands and release to rice seedlings (Fig. 5a, b). This indicates that NcVg facilitates the release of RDV to rice plants. The dsNcVg-treated viruliferous leafhoppers feeding cause limited change in JA- or SA-related gene expression compared to dsGFP-treated viruliferous leafhoppers feeding (Supplementary Fig. 6). However, dsNcVg-treated viruliferous leafhoppers feeding led to an increased production of $H_2O_2$, the activity of CAT and POD, and accumulation of MDA in rice seedlings (Fig. 5c). DAB and H2DCFDA staining assays showed that dsNcVg-treated viruliferous leafhoppers feeding caused a higher number of

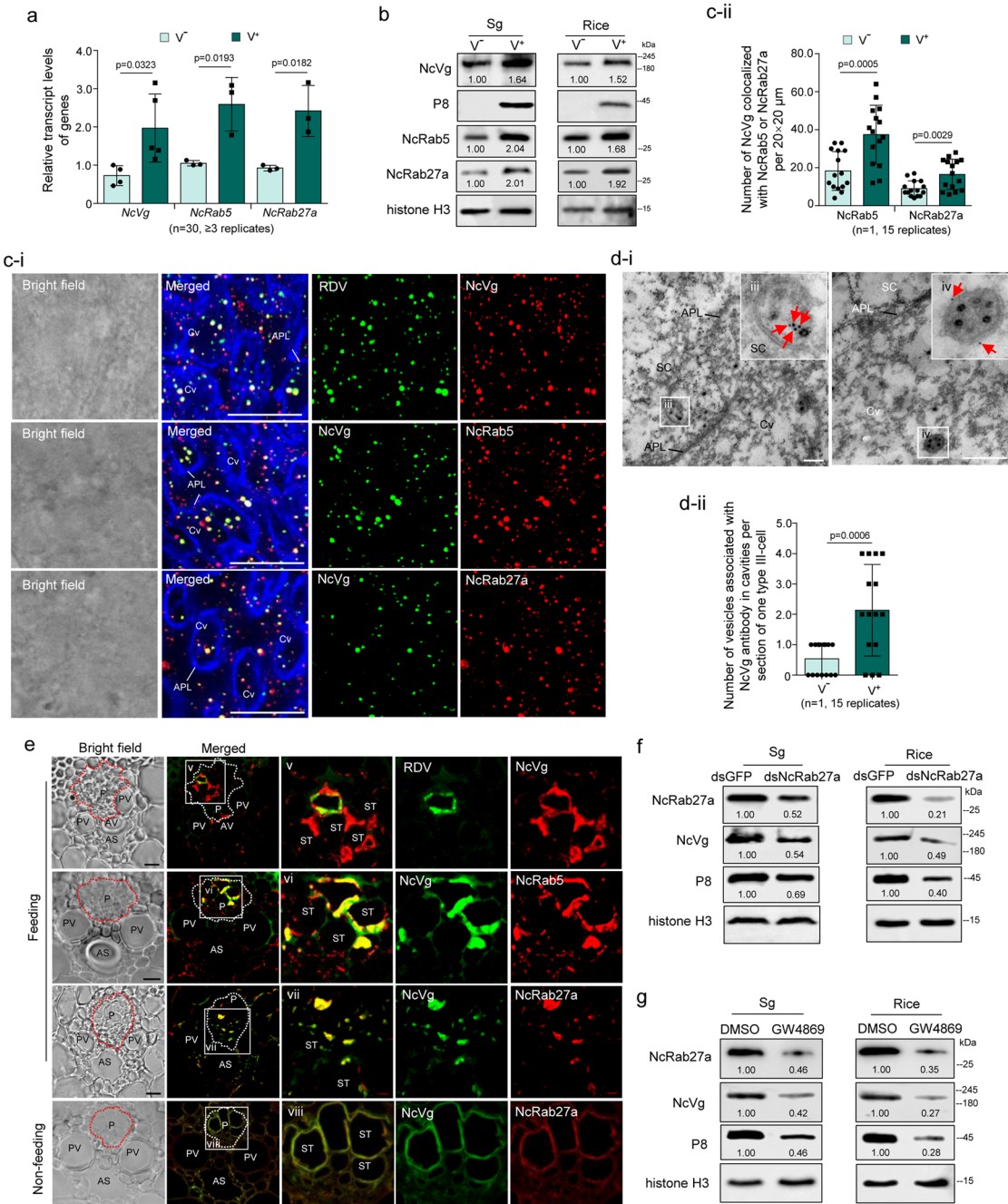

**Fig. 2 | RDV hijacking NcVg then releasing together from salivary glands to rice phloem via exosomal pathway. a**, **b** RDV infection inducing NcVg, NcRab5, and NcRab27a expression in salivary glands and release to rice plants, as determined by RT-qPCR **a** and western blot **b** assays. **c** RDV colocalizing with NcVg and inducing colocalization of NcVg with NcRab5 or NcRab27a in cavities and cytoplasm of salivary gland, as determined by immunofluorescence microscopy. APL, apical plasmalemma. Cv, cavity. Bars, 10 μm. The mean number of NcVg colocalizing with NcRab5 or NcRab27a in uninfected or infected salivary gland are shown in **c-ii**. Fifteen random 20 × 20 μm fields of samples from infected or uninfected salivary glands were examined. V + , infected salivary glands. V-, uninfected salivary glands. **d** Immunoelectron microscopy showing the localization of NcVg at RDV-packaging exosomes within the cytoplasm and cavities of salivary glands. Salivary glands of leafhoppers were immunolabeled with NcVg-specific IgG as the primary antibody, followed by treatment with 15-nm gold particle-conjugated IgG as the secondary

antibody. Panels iii to iv are the enlarged images of the boxed areas in **d-i**. Red arrows indicate gold particles. Bar, 200 nm. The mean number of vesicles associated with NcVg antibody in cavities per section of one type III-cell of one uninfected or infected salivary gland are shown in **d-ii**. Fifteen random samples from infected or uninfected salivary glands were examined. **e** Colocalization of NcVg with RDV, NcRab5, and NcRab27a in rice phloem, as determined by immunofluorescence microscopy. Panels v to viii are the enlarged images of the boxed areas in **e**. P, phloem; AS, air space; PV, pitted vessel; ST, sieve tube. Bars, 10 μm. **f**, **g** Knockdown of *NcRab27a* expression **f** or treatment of GW4869 **g** reducing NcVg and NcRab27a accumulation in and release from salivary glands of nonviruliferous leafhoppers. Relative intensities of bands in western blot assays are shown. Data in **a**, **b**, **c-ii**, **e**, **d-ii**, **f** and **g** represent at least 3 biological replicates. Means ( ± SD) in **a**, **c-ii** and **d-ii** are analyzed using two-tailed *t*-test.

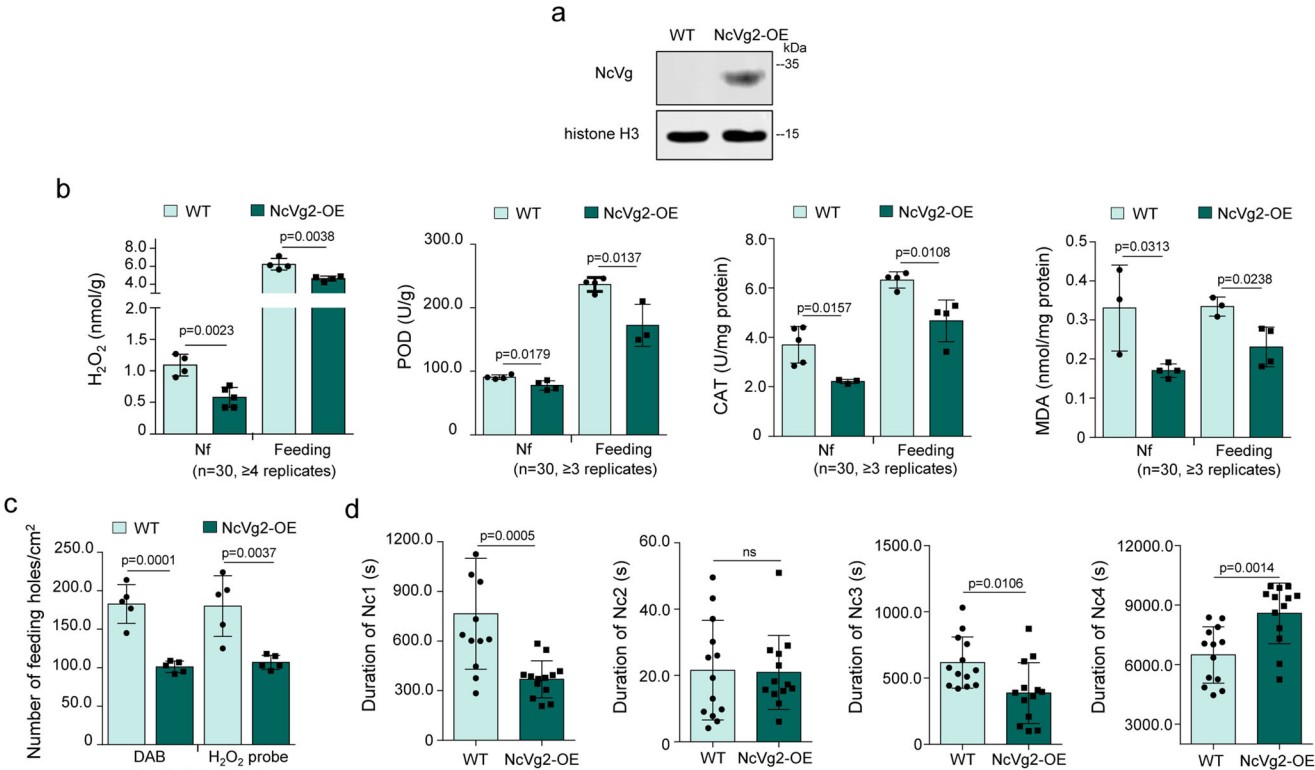

**Fig. 3 | Overexpression of NcVg2 reducing $H_2O_2$ burst and promoting leafhopper feeding. a** NcVg2 expression in NcVg2-OE lines, as determined by western blot assays. Proteins from NcVg2-OE lines or WT were detected using NcVg- or histone H3-specific antibody in western blot assays. **b** Overexpression of NcVg2 reducing $H_2O_2$ burst and metabolism in rice plants exposed or not exposed to leafhoppers, as determined by content of $H_2O_2$ and MDA, as well as CAT and POD activity. Data are shown from 1 rice seedlings exposed or not exposed to 30 nonviruliferous leafhoppers. **c** The number of feeding holes per cm² of leaves of WT or

NcVg2-OE plants exposed to leafhoppers, as determined by DAB or H2DCFDA staining. One leaf of rice seedling of WT or NcVg2-OE plants exposed to 5 nonviruliferous leafhoppers for 12 h was tested. **d** Overexpression of NcVg2 promoting plant penetration behaviour of leafhopper, as determined by EPG technique. Each nonviruliferous leafhopper was continuously and electrically recorded during 3-hour feeding periods. Data in **a**, **b** and **c** represent at least 3 biological replicates. Data in **d** represent 13 valid biological replicates. Means ( ± SD) in **b, c** and **d** are analyzed using two-tailed *t*-test. Ns, not significant.

feeding holes compared to dsGFP-treated viruliferous leafhoppers (Fig. 5d). EPG assays demonstrated that knocking down *NcVg* expression in viruliferous leafhoppers significantly prolonged the duration of non-probing (Nc1) and salivation (Nc3), but shortened the duration of ingestion from phloem and xylem bundle tissues (Nc4), compared with dsGFP-treated viruliferous leafhoppers (Fig. 5e). This suggests that the dsNcVg-treated viruliferous leafhoppers encountered barriers during feeding. The dsNcVg-treated viruliferous leafhoppers also caused lower transmission rate of RDV than dsGFP treatments (Fig. 5f). Taken together, these results revealed that reduced NcVg release promotes the production of $H_2O_2$, which is disadvantage for leafhopper feeding, finally blocking RDV transmission.

### NcVg interacts with OsGSTF12 to suppress $H_2O_2$ bursts in rice plants

To determine whether the mechanism of NcVg effector in leafhoppers suppressing the production of $H_2O_2$ was similar to small planthoppers[24], the interaction of NcVg with OsWRKY71 was examined. However, it was found that NcVg1 to NcVg4 were unable to interact with OsWRKY71 in the Y2H system (Supplementary Fig. 7a). Therefore, a Y2H system was used to screen rice candidates that interact with NcVg2 from a cDNA library of *Oryza sativa* L.ssp. *Japonica* cv. Nipponbare. The candidate that attracted attention was OsGSTF12, because GST enzymes can catalyze glutathione (GSH)-dependent oxidation reactions that scavenge excess amounts of $H_2O_2$[34–38]. GSH is then converted into glutathione disulphide (GSSG), an oxidized form of GSH (Fig. 6a). OsGSTF12 was found to possess a domain of the

Glutathione S-transferase superfamily (Supplementary Fig. 7b), suggesting that OsGSTF12 probably had the ability to catalyze GSH-dependent oxidation reaction. Therefore, OsGSTF12 was identified as the potential interactor of NcVg2 and was chosen for further analysis. Both Y2H and bimolecular fluorescence complementation (BiFC) assays demonstrated the specific interaction of NcVg and OsGSTF12 (Fig. 6b, c). Immunofluorescence assays showed the colocalization of NcVg2 and OsGSTF12 in the phloem of rice leaves that were exposed to leafhoppers (Fig. 6d). These results suggest an interaction between NcVg2 and OsGSTF12 in vivo or in vitro, and that NcVg recruits OsGSTF12 in rice phloem when leafhoppers feed.

The biological function of OsGSTF12 during leafhopper feeding was then investigated. Transient expression of OsGSTF12 in *Nicotiana benthamiana* significantly improved the activity of GST and the production of GSSG, but reduced the accumulation of $H_2O_2$ and GSH, compared to transient expression of GFP (Fig. 6e, f). These results indicated that OsGSTF12 expressed in *N. benthamiana* had the ability to scavenge excess $H_2O_2$ by catalyzing GSH to generate GSSG. Further investigation showed that leafhoppers feeding on rice seedlings for 12 h caused a significant increase in OsGSTF12, as well as the activity of GST, accumulation of GSH, and production of GSSG, compared to rice seedlings that were not exposed to leafhoppers (Fig. 6g, h). Rice seedlings exposed to viruliferous leafhoppers showed higher accumulation of OsGSTF12, activity of GST, and production of GSSG, while having lower content of GSH in rice plants, compared with rice seedlings exposed to nonviruliferous leafhoppers (Fig. 6i, j). Additionally, knocking down *NcVg* expression in nonviruliferous or viruliferous leafhoppers caused a significant decrease in OsGSTF12 in rice seedlings

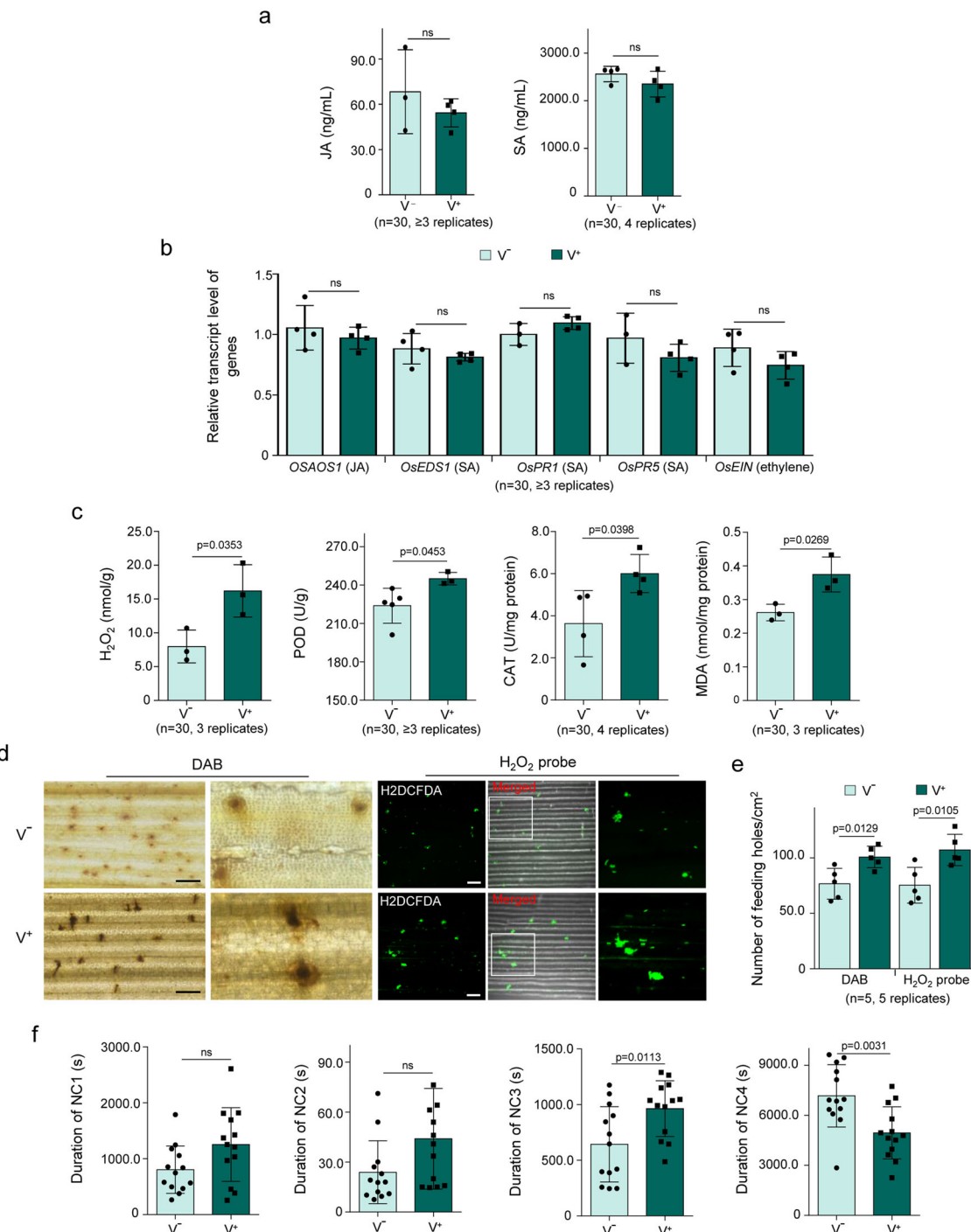

**Fig. 4 | Exposure to viruliferous leafhoppers for 12 h triggering $H_2O_2$ burst in rice plants. a**, **b** Exposure to viruliferous leafhoppers for 12 h causing limited effect on the content of JA and SA **a** as well as the expression of genes related to JA, SA, and ethylene **b**, as determined by mass spectrometer and RT-qPCR assays. **c** Exposure to viruliferous leafhoppers inducing $H_2O_2$ burst and metabolism in rice plants, as determined by the content of $H_2O_2$ and MDA, as well as the activity of CAT and POD. Data in **a**–**c** are shown from 1 rice seedling exposed to 30 nonviruliferous or viruliferous leafhoppers. **d** and **e** Exposure to viruliferous leafhoppers increasing $H_2O_2$ accumulation at feeding holes, as determined by DAB or H2DCFDA staining. One leaf of a rice seedling exposed to 5 nonviruliferous or viruliferous leafhoppers for 12 h was tested. The mean number of feeding holes per cm² of leaves are shown in **e**. Bars, 200 μm. **f** Viruliferous leafhoppers encountering difficulty in feeding, as determined by EPG technique. Each nonviruliferous or viruliferous leafhopper was continuously and electrically recorded during 3-hour feeding periods. V + , viruliferous. V-, nonviruliferous. Data in **a**, **b**, **c** and **e** represent at least 3 biological replicates. Data in **f** represent 13 valid biological replicates. Means ( ± SD) in **a**, **b, c, e,** and **f** are analyzed using two-tailed *t*-test. Ns, not significant.

exposed to leafhoppers for 12 h, as well as the activity of GST, accumulation of GSH, and production of GSSG, compared to dsGFP treatment (Fig. 6k and l). These results indicate that NcVg of nonviruliferous and viruliferous leafhoppers suppresses $H_2O_2$ production via its interaction with OsGSTF12.

Using the CRISPR/Cas9 technology, OsGSTF12-knockout (KO) transgenic plants were generated (Supplementary Fig. 8a). Sequences of *OsGSTF12* gene of OsGSTF12-KO plants showed the substitution or deletion in different number of nucleic acids (Supplementary Fig. 8b). These OsGSTF12-KO plants displayed varying activities of GST

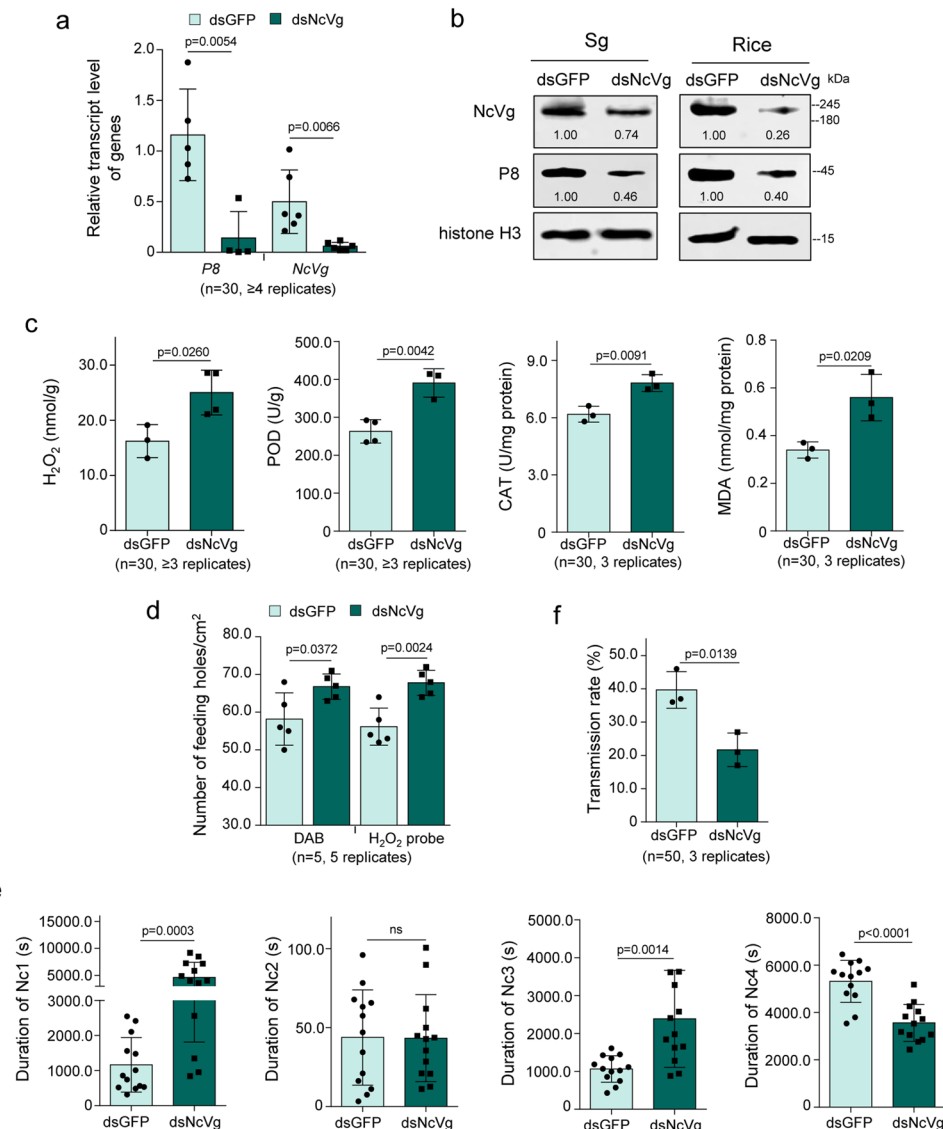

**Fig. 5 | RDV exploiting NcVg effector to suppress $H_2O_2$ burst of rice plants for facilitating transmission. a**, **b** Knockdown of *NcVg* expression in leafhoppers reducing RDV accumulation in salivary glands and release to plants, as determined by RT-qPCR **a** and western blot **b** assays. Data in **a** are shown from salivary glands of 30 dsGFP- or dsNcVg-treated viruliferous leafhoppers. The proteins were detected using NcVg-, P8-, or histone H3-specific antibodies in western blot assays, and relative intensities of bands are shown. **c** Knockdown of *NcVg* expression in viruliferous leafhoppers enhancing $H_2O_2$ burst and metabolism in rice plants, as determined by content of $H_2O_2$ and MDA, as well as activity of CAT and POD. Data are shown from 1 rice seedling exposed to 30 dsNcVg- or dsGFP-treated viruliferous leafhoppers. **d** Knockdown of *NcVg* expression in viruliferous leafhoppers increasing the number of feeding holes, as determined by DAB or H2DCFDA staining. Data are shown from 1 leaf of a rice seedling exposed to 5 dsNcVg- or dsGFP-treated viruliferous leafhoppers for 12 h. **e** Knockdown of *NcVg* expression disadvantageous for viruliferous leafhoppers feeding, as determined by EPG technique. Each dsNcVg- or dsGFP-treated viruliferous leafhopper was continuously and electrically recorded during a 3-hour feeding period. Means ($\pm$ SD) are shown and represent 13 valid biological replicates. **f** Knockdown of *NcVg* expression in viruliferous leafhoppers reducing the RDV transmission rate. Means ($\pm$ SD) are shown from 50 dsNcVg- or dsGFP-treated viruliferous leafhoppers individually feeding on 1 rice seedling. Data in **a**, **b**, **c**, **d**, and **f** represent at least 3 biological replicates. Means ($\pm$ SD) in **a**, **c**, **d**, **e** and **f** are analyzed using two-tailed *t*-test. Ns, not significant.

(Supplementary Fig. 8c). For further analyses, lines #12 and #29 with the lowest activity of GST were chosen (Supplementary Figs. 8c–e).

As Fig. 7a shows, either OsGSTF12-KO plants exposed or not exposed to leafhoppers showed lower activity of GST and production of GSSG, while higher content of GSH in rice plants. These OsGSTF12-KO plants also demonstrated higher $H_2O_2$ accumulation, as well as activity of CAT and POD, and accumulation of MDA, compared to the WT (Fig. 7b). DAB and H2DCFDA staining assays revealed that leafhoppers feeding caused a higher number of feeding holes in OsGSTF12-KO plants than the WT (Fig. 7c). Additionally, EPG assays showed that leafhoppers feeding on OsGSTF12-KO plants took a significantly longer duration of non-probing (Nc1) and salivation (Nc3) but a shorter duration of ingestion from phloem and xylem bundle

tissues (Nc4) (Fig. 7d). These findings revealed that OsGSTF12-KO plants are adverse to leafhoppers feeding due to the high accumulation of $H_2O_2$. Moreover, NcVg2-OE plants exposed or not exposed to leafhoppers demonstrated a higher level of OsGSTF12, activity of GST, and production of GSSG, as well as a low level of accumulation of GSH, compared to the WT (Fig. 7e, f). These results indicate that the overexpression of NcVg induces more OsGSTF12 to scavenge $H_2O_2$, ultimately suppressing the production of $H_2O_2$.

## Overexpression of NcVg enhances RDV transmission by leafhoppers

The effects of NcVg2-OE and GSTF12-KO plants on RDV transmission by leafhoppers were analyzed. Western blot assays demonstrated the

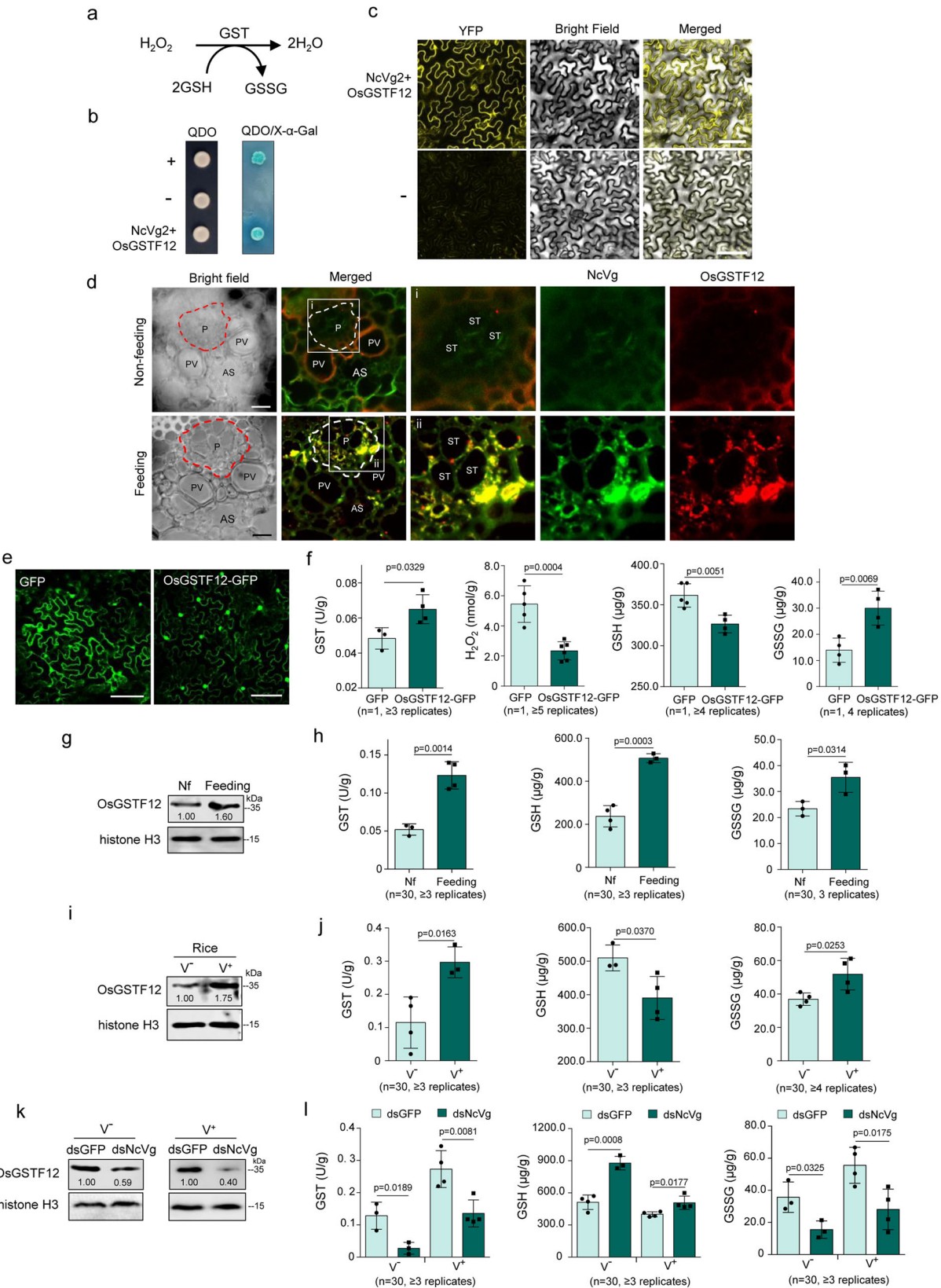

highest accumulation of P8 in NcVg2-OE plants exposed to viruliferous leafhoppers but the lowest accumulation of P8 in GSTF12-KO plants (Fig. 8a). This finding suggests that overexpression of NcVg in rice plants facilitates RDV release, while knockout of GSTF12-KO adversely affects RDV release. Furthermore, the production of $H_2O_2$, activity of CAT and POD, as well as accumulation of MDA, resulted from

viruliferous leafhopper feeding on NcVg2-OE plants were the lowest, while those on GSTF12-KO plants were the highest (Fig. 8b). DAB and H2DCFDA staining assays revealed that viruliferous leafhopper feeding on NcVg2-OE and GSTF12-KO plants respectively caused the lowest and highest number of feeding holes (Fig. 8c). The activity of GST and production of GSSG caused by viruliferous leafhopper feeding on

**Fig. 6 | NcVg interacting with OsGSTF12 to inhibit H$_2$O$_2$ burst. a** GST catalyzing the reaction of GSH and consuming H$_2$O$_2$. **b** Y2H assays showing the interaction of NcVg2 with OsGSTF12. +, positive control; −, negative control. QDO, SD/-Trp-Leu-His-Ade medium. **c** BiFC assays showing the interaction of NcVg2 with OsGSTF12 in *N. benthamiana* leaf cells. −, negative control. Bars, 100 μm. **d** Immunofluorescence microscopy demonstrating the colocalization of NcVg and OsGSTF12 in rice phloem exposed to leafhoppers. Panels i to ii (merged) are enlarged images of the boxed areas in **d**. AS, air space; P, phloem; PV, pitted vessel; ST, sieve tube. Bars, 10 μm. **e, f** The transient expression of OsGSTF12 increasing H$_2$O$_2$ accumulation and activating GST-catalyzing reactions in *N. benthamiana*. The leaves transiently expressing OsGSTF12 **e** were tested for GST activity and GSH and GSSG content **f**. Bars, 200 μm. **g** OsGSTF12 expression in rice seedlings induced by leafhopper feeding, as determined by western blot assays. OsGSTF12- and histone H3-specific antibodies were used to detect proteins in western blot assays. **h** GST-catalyzing

reactions in rice seedlings induced by leafhopper feeding, as determined by GST activity and GSH and GSSG content. Nf, non-feeding. **i, j** Exposure to viruliferous leafhopper improving OsGSTF12 expression and promoting GST catalyzing reaction, as determined by western blot **i**, activity of GST, contents of GSH and GSSG **j**. The proteins were detected by using OsGSTF12- or histone H3-specific antibody in western blot assays. Data in **h, j,** and **l** are shown from 1 rice seedling exposed or not exposed to 30 nonviruliferous or viruliferous leafhoppers. **k, l** Knockdown of *NcVg* expression in nonviruliferous or viruliferous leafhoppers decreasing OsGSTF12 expression and GST-catalyzing reactions. Western blot assays were used to detect proteins using OsGSTF12- and histone H3-specific antibodies. Data in **f, g, h, i, j, k** and **l** represent at least 3 biological replicates. Means (± SD) in **c, d, f, h, j** and **l** are shown and analyzed using two-tailed *t*-test. V-, nonviruliferous leafhoppers, V + , viruliferous leafhoppers.

NcVg2-OE plants were the highest, while the accumulation of GSH were the lowest (Fig. 8d). Viral transmission tests indicated that exposure of NcVg2-OE plants resulted in the highest transmission rate of RDV (Fig. 8e). Taken together, overexpression of NcVg2 in rice plants enhances RDV transmission effect on RDV transmission.

To understand whether NcVg2 had an inherent ability to suppress H$_2$O$_2$ by inducing GST activity, the domains of NcVg1 to NcVg4 were expressed separately in *N. benthamiana* (Supplementary Fig. 9a). It was found that leaves transiently expressing NcVg2 had the lowest content of H$_2$O$_2$ and GSH, and the highest activity of GST, compared to those expressing NcVg1, NcVg3, or NcVg4 (Fig. 8f). Then the stem bases of rice seedlings were separately injected with prokaryotically expressed NcVg1 to NcVg4 in equal amounts (Supplementary Fig. 9b). The results showed that at 1-hour post injection, injection with NcVg2 caused significantly lower level of H$_2$O$_2$, and GSH in rice seedlings while higher activity of GST and content of GSSG, compared to injection with NcVg1, NcVg3, or NcVg4 (Fig. 8g). These results illustrated that NcVg2 has an inherent ability to target GST and induce GST activity, which catalyzes GSH-dependent oxidation, then suppresses H$_2$O$_2$ accumulation.

## Discussion

Herbivores feeding can activate JA, JA-Ile, SA, ET, and ROS burst of plants. ROS, particularly H$_2$O$_2$, plays a crucial role in herbivore-associated molecular patterns-triggered immunity[39,40]. Plants can enhance their insect resistance by regulating the H$_2$O$_2$ pathway[41–44]. Our study showed that the defense of rice exposed to leafhoppers for 12 h was dominated by H$_2$O$_2$ burst. The salivary NcVg induced the expression and activity of OsGSTF12, which suppressed H$_2$O$_2$ burst, and ultimately benefiting leafhopper feeding. In contrast, at the early stage of viral transmission by insects, when virus delivered to plants has not yet replicated or spread, the plant defense mainly associates with insect-resistance, rather than virus-resistance. We found that RDV infection in leafhoppers increased H$_2$O$_2$ burst of rice and made it more difficult for the leafhoppers to feed, indicating an enhancement in plant defense to insects. It is believed that RDV infection triggers the production of various salivary effectors and elicitors, which trigger the plant defense. During this process, RDV-upregulated NcVg induced more OsGSTF12 through direct interaction to suppress H$_2$O$_2$ burst. In other words, the OsGSTF12, acting as an interactor, responds to NcVg. We also found that dsNcVg treated-leafhoppers also took a long time to salivate. One possible explanation is that when NcVg was knock-down, leafhoppers had to produce more salivary effectors to suppress ROS burst, consequently prolonging the duration of salivation. Therefore, the individual effect of NcVg on rice defense is not equivalent to the integrated effects of RDV infection on rice defense. There is no contradiction between the increase in H$_2$O$_2$ levels and the higher expression of OsGSTF12 caused by exposure to viruliferous leafhoppers. RDV inducing more salivary NcVg to promote transmission by leafhoppers, allows the RDV to accumulate and prepare better

for the subsequent viral propagation and spread. We anticipate that this is an example of how virus improving counter defense of insect to plant for transmission as insect feed.

Vg has been acknowledged as crucial in providing nutrients for oocytes, regulating immune defense, and facilitating the vertical transmission of viruses across insect generations[20–22,45,46]. In contrast, research on the salivary Vg of piercing-sucking insects has been limited[24,25], especially the pathway of Vg release from the salivary glands and its function post-release. Regarding the different modes of NcVg release from salivary glands of viruliferous and nonviruliferous leafhoppers, it was postulated that the proportion of exosomal NcVg was the key reason. Our previous study revealed a low abundance of exosomes in the cavities of uninfected salivary glands, whereas RDV infection induces a significant increase in the number of exosomes[17]. In this study, only a small fraction of NcVg antigens were within exosomes in cavities of uninfected salivary glands. This suggests that the low proportion of exosomal NcVg is likely attributed to the limited number of exosomes. In contrast, in the RDV-infected salivary glands, the number of exosome increased and NcVg entered the virus-induced exosomes through the interaction between NcVg2 and NcRab5. Consequently, this led to a higher proportion of exosomal NcVg. Putative interactors of NcVg2, such as vesicle-associated membrane protein, synaptobrevin, rabankyrin that is a Rab5 effector, in Y2H screening can be further studied, because they also associate with vesicle biogenesis and trafficking. Vg has been reported to be present in extracellular vesicles of murine whipworm[47], indicating a close association of Vg with vesicles. Further research is needed to investigate whether saliva Vg of most piercing-sucking insects is possibly implicated in exosomes for release. Additionally, the 220-kDa NcVg is synthesized by the fat body[26], but was detected in the salivary glands and rice plants in this study. It is meaningful to investigate whether this 220-kDa NcVg is transported from the fat body to salivary glands or synthesized by the salivary glands.

Plant GSTs (EC 2.5.1.18; GSTs) utilize GSH as a co-substrate or coenzyme to catalyze a variety of reactions, including peroxidase reactions[36,48]. The GSH-dependent peroxidase activities of GSTs can scavenge toxic hydroperoxides to protect from ROS, oxidative damage and maintain cellular redox homeostasis[37,49]. These GST enzymes use GSH as an electron donor to reduce hydroperoxides and generate GSSG[37]. Plant GSTs are a diverse family of multifunctional enzymes, and categorized into distinct classes, including phi (GSTF), tau (GSTU), zeta (GSTZ), and so on[50]. GSTF12 play a crucial role in anthocyanin accumulation in cotton (*Gossypium hirsutum* L.) and flavonoid transport to the vacuole in *A. thaliana*[49,51]. In this study, OsGSTF12 contained the conserved domain of GST and displayed significant GST activity when was expressed in *N. benthamiana*. As the target protein of NcVg, OsGSTF12 was induced by NcVg and showed improved expression and activity, promoting GSH substrate to react with H$_2$O$_2$, resulting in reduced H$_2$O$_2$ burst. These results confirm that GSTF12 of *O. sativa* also possesses GST activity and is exploited in the process of insect defense against plant hosts.

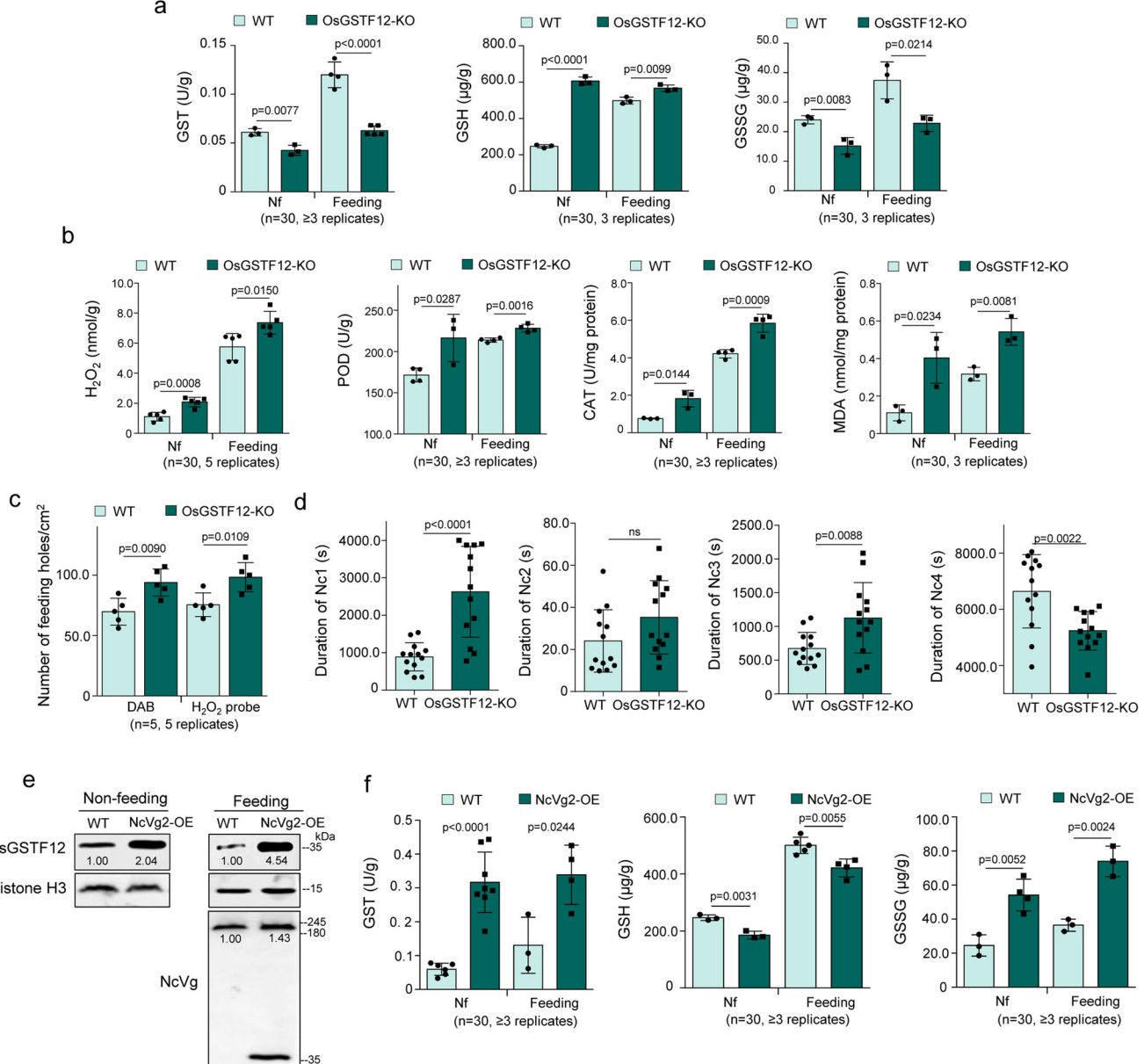

**Fig. 7 | Knockout of OsGSTF12 promoting H₂O₂ burst and suppressing leaf-hopper feeding. a** Knockout of *OsGSTF12* reducing GST catalyzing reaction in rice plants exposed or not exposed to leafhoppers, as demonstrated by GST activity and contents of GSH and GSSG. **b** Knockout of *OsGSTF12* increasing H₂O₂ burst and metabolism in rice plants exposed or not exposed to leafhoppers, as indicated by content of H₂O₂ and MDA, as well as activity of CAT and POD. **c** Knockout of *OsGSTF12* increasing the number of feeding holes in leaves exposed to leafhoppers, as detected by DAB or H2DCFDA staining. One leaf of rice seedling of WT or OsGSTF12-KO lines exposed to 5 nonviruliferous leafhoppers for 12 h was tested. **d** Knockout of *OsGSTF12* increasing difficulty of leafhoppers feeding, as determined by EPG technique. Each nonviruliferous leafhopper was continuously and electrically recorded during a 3-hour feeding period. Means (± SD) represent 13 valid biological replicates. **e** Overexpression of NcVg2 increasing OsGSTF12 expression in rice plants exposed or not exposed to leafhoppers, as determined by western blot assays using OsGSTF12-, NcVg-, or histone H3-specific antibody. **f** Overexpression of NcVg2 promoting GST catalyzing reaction in rice plants exposed or not exposed to leafhoppers, as demonstrated by GST activity and contents of GSH and GSSG. Data in **a**, **e**, and **f** are shown from 1 rice seedling exposed or not exposed to 30 nonviruliferous leafhoppers. Means (± SD) in **a**–**d** and **f** represent at least 3 replicates, and are analyzed using two-tailed *t*-test. Ns, not significant.

Exosomes, derived from the endosomal system, carry inter-cellular materials for information transfer and material exchange[52]. They deliver immune signals that resist viruses[53], viral proteins that facilitate viral infection[54,55], viral nucleic acids that assist in viral infection, and virions for intercellular spread[56,57]. Plant exosomes can also carry virus-induced small RNAs to inhibit pathogen infection[58]. Exosomes of planthopper and *Drosophila* respectively transport virus to plant hosts and carry virus-induced small interfering RNAs to trigger antiviral immunity[59–61]. Therefore, exosomes likely play an essential role in insect-microbe or insect-microbe-host interactions by safely delivering biological factors to interacting organisms.

Rabs are the key regulators of intercellular vesicle budding, trafficking, and fusion, even exosome formation and trafficking[62,63]. Several Rabs are implicated in the biogenesis and release of exosomes, including Rab27 and Rab5[58,64,65]. Rab5 generally localizes to early endosomes where they drive endosome trafficking, and is within the exosomes[63]. Rab27 regulates the fusion of late endosomes at the plasma membrane and functions in the docking site to release

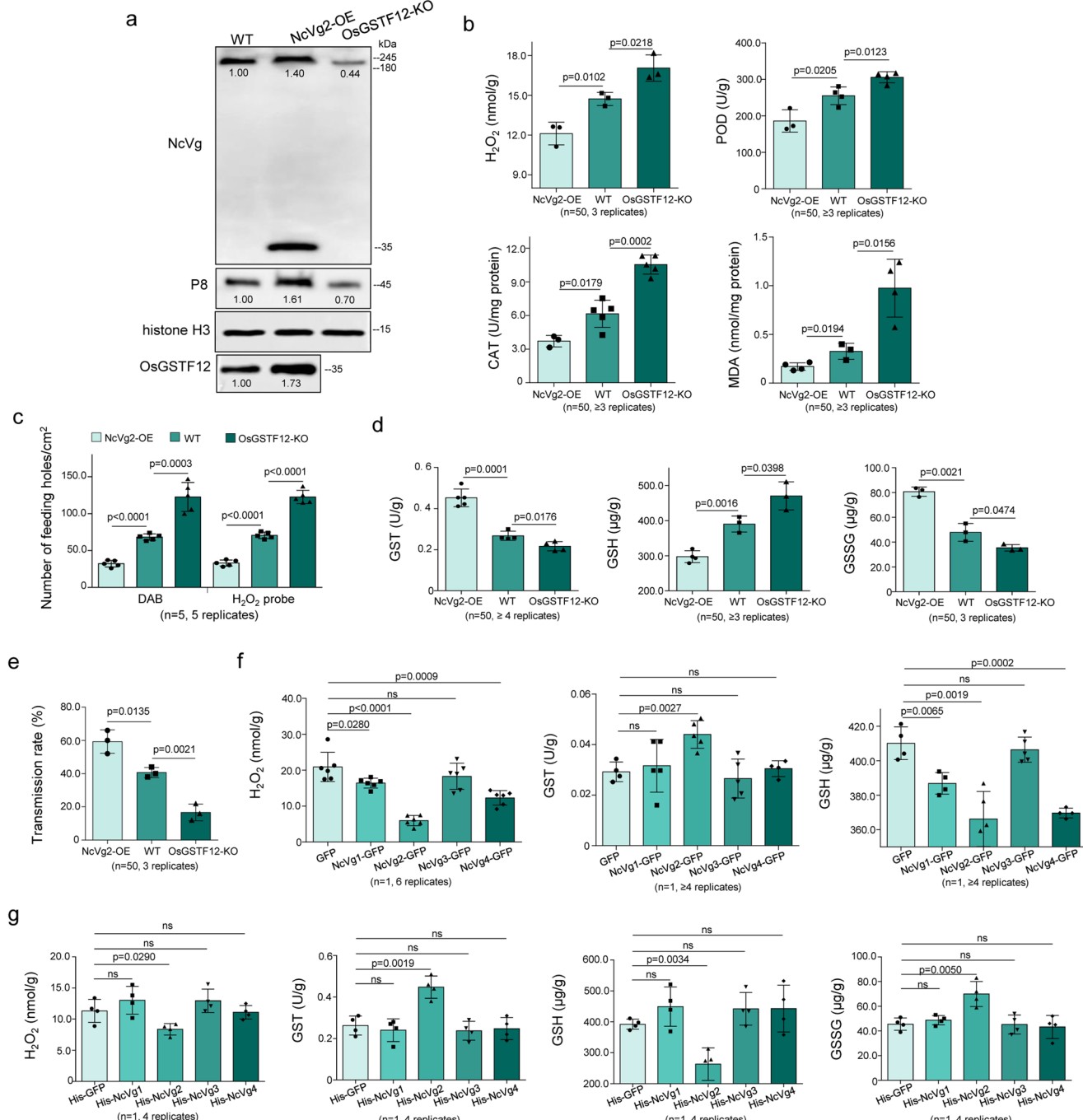

**Fig. 8 | Overexpression of *NcVg2* promoting RDV transmission by leafhoppers via suppressing H₂O₂ burst of rice plants. a** Overexpression of *NcVg2* promoting RDV release, while knockout of *OsGSTF12* inhibiting RDV release, as indicated by western blot assays using NcVg-, P8-, OsGSTF12-, or histone H3-specific antibody. The relative intensity of the bands is shown. **b** Viruliferous leafhoppers feeding caused the lowest levels of H₂O₂ burst and metabolism in NcVg2-OE lines compared to WT or OsGSTF12-KO lines. **c** Exposure of NcVg2-OE lines to viruliferous leafhoppers causing the lowest number of feeding holes compared to feeding on WT or OsGSTF12-KO lines, as detected by DAB or H2DCFDA staining, respectively. One leaf of rice seedling of WT, NcVg2-OE, or OsGSTF12-KO lines exposed to 5 viruliferous leafhoppers for 12 h was tested. **d** Exposure of NcVg2-OE lines to viruliferous leafhoppers causing the most active GST catalyzing reaction in NcVg2-OE

lines, compared to WT or OsGSTF12-KO lines, as demonstrated by GST activity, GSH and GSSG contents. Data are shown from 1 leaf of WT, NcVg2-OE, or OsGSTF12-KO lines exposed to 50 viruliferous leafhoppers. **e** NcVg2-OE lines beneficial to RDV transmission by leafhoppers, as indicated by the transmission rate. Data are shown from 50 viruliferous leafhoppers individually feeding on 1 rice seedling of WT, NcVg2-OE, or OsGSTF12-KO lines. **f** The transient expression of NcVg2 significantly suppressing H₂O₂ accumulation, inducing GST activity and reducing GSH content of *N. benthamiana*. The leaves separately expressed NcVg1 to NcVg4 were tested for the H₂O₂ contents, GST activity and GSH content. **g** Injection of rice seedlings with His-Vg2 significantly reducing contents of H₂O₂, and increasing GST-catalyzing reaction in rice plants. All data represent at least 3 biological replicates. Means (± SD) in **b, c, d, e, f**, and **g** are analyzed using two-tailed *t*-test. Ns, not significant.

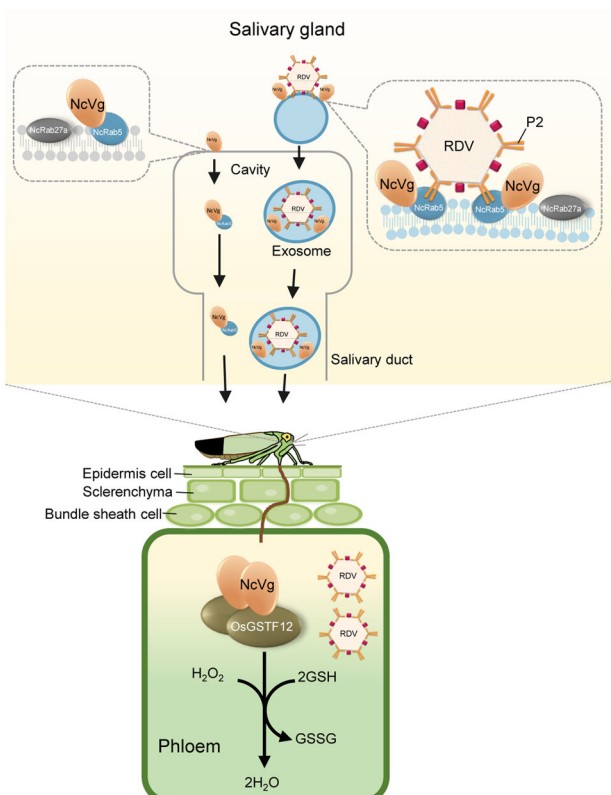

**Fig. 9 | Proposed model of RDV hijacking NcVg for transmission.** NcVg of nonviruliferous leafhoppers associates with NcRab5 via interaction, facilitating their release together from salivary glands to cavities and finally entering rice phloem when leafhoppers feed. The released NcVg in rice phloem induces the expression and activity of OsGSTF12, which catalyzes GSH-dependent oxidation, ultimately scavenging excessive $H_2O_2$ and benefiting leafhopper feeding. The RDV infection in salivary glands indirectly induces NcVg accumulation mediated by RDV-upregulated NcRab5, resulting in the formation of RDV-NcVg-NcRab5 complexes. The induced complexes are packaged into RDV-induced exosomes and then released into cavities, eventually entering rice phloem. The upregulated NcVg in rice phloem enhances the GSH-dependent oxidation catalysis and $H_2O_2$ scavenging, finally facilitating leafhoppers feeding and viral transmission.

exosomes[62]. Many viruses exploit Rabs to their advantage, such as hepatitis C virus and Herpes simplex virus-1[66,67]. Previously, we discovered that RDV exploits the interaction between P2 and NcRab5 in the exosomal pathway to hijack exosomes for release into the salivary cavity via a NcRab27a-dependent pathway[17]. Subsequently, virus-packaging exosomes enter the rice phloem for viral delivery. In this study, we further uncovered that RDV is associated with NcVg via NcRab5 and form the RDV-NcRab5-NcVg complexes, which were packaged in exosomes, finally reaching the rice phloem to modulate rice defense. This exosome-mediated release of the virus and virus-induced effectors reduces damage to salivary glands and facilitates persistent viral transmission by insects. Furthermore, exosomes prevent the virus from being attacked by the insect immune system, ensuring the establishment of initial infection for the virus in the phloem. The exosomal pathway is likely a conserved strategy among plant arboviruses and effectors that are essential for the release of cross-kingdom biological factors facilitating viral transmission to the plant host. This study provides evidence of viruses utilizing insect exosomes to deliver virus-induced biological factors, thus facilitating viral transmission. In addition, it showed partial colocalization of NcRab5 and NcVg in salivary glands and feeding sites of rice plants (Fig. 1e). It is postulated that NcRab5 also functions in other mechanisms independent of NcVg.

Therefore, we propose a model illustrating how RDV exploits the NcVg effector for transmission (Fig. 9). NcVg associates with NcRab5 through interaction for release together from salivary glands to cavities and eventually reaches the rice phloem during leafhoppers feed. Once released into the rice phloem, NcVg induces the expression and activity of OsGSTF12, which catalyzes the conversion of $H_2O_2$ to $H_2O$, thus facilitating leafhopper feeding. RDV in salivary glands induces and hijacks NcVg to virus-induced exosomes through NcRab5, and releases them together in rice plants. Upon reaching the phloem, upregulated NcVg induces OsGSTF12 expression and activity, resulting in extensive $H_2O_2$ scavenging, which ultimately facilitates leafhopper feeding and viral transmission.

## Methods
### Insects, viruses and antibodies
Nonviruliferous individuals of *N. cincticeps* were collected from rice fields in Fujian Province, southeastern China, and propagated for several generations in the laboratory. The initial source of RDV-infected rice plants was also collected from rice fields in Fujian Province and propagated via transmission by *N. cincticeps* under greenhouse conditions.

Dr. Toshihiro Omura of the National Agricultural Research Center, Japan, provided rabbit polyclonal antisera against RDV antigens. Polyclonal antibodies against P8 were obtained from ABclonal, China, while polyclonal antibodies against NcRab27a and NcRab5 were sourced from Beyotime, China. Genscript Biotech Corporation, Nanjing, China, prepared polyclonal antibodies against NcVg and OsGSTF12, and the process was approved by the Science Technology Department of Jiangsu Province of China.

For the preparation of conjugates, IgGs against NcVg or RDV antigens were directly conjugated to FITC to generate NcVg- or RDV-FITC. Meanwhile, IgGs against NcVg, OsGSTF12, NcRab27a or NcRab5 were directly conjugated to rhodamine to generate NcVg-, OsGSTF12-, NcRab27a- or NcRab5-rhodamine, following the manufacturer's instructions of Thermo Fisher Scientific. The actin dyes Phalloidin-Alexa Fluor 647 carboxylic acid were obtained from Thermo Fisher Scientific (A22287). Mouse monoclonal antibodies against 6×His tag, GST, and GFP were purchased from Transgene Biotech (HT501, HT601, and HT801).

### Western blots detecting target proteins in salivary glands and rice plants
Protein was extracted in equal amounts from the ovaries of 30 female leafhoppers to examine the cleavage pattern of NcVg. Protein was extracted from the whole bodies or salivary glands of 30 non-viruliferous or viruliferous leafhoppers to examine the target proteins. Equal amounts of protein were loaded for western blot assay, with this experiment performed in triplicate and repeated 3 times for each treatment.

To investigate the release of salivary proteins in rice plants, approximately 50 adult leafhoppers were starved for 2 h before feeding on a single rice seedling at the 2-leaf stage (approximately 10 cm in height) for 24 h. The tested plants were then collected to extract equal quantities of protein.

### Interaction of NcVg with NcRab5 of leafhoppers
For the examination of the interaction between NcVg and NcRab5 of leafhoppers, total RNAs of *N. cincticeps* were isolated for a cDNA library construction by Oebiotech company of China. The cDNA library was constructed in pGADT7 vector (Clontech, K1612−1) for screening putative interactors of NcVg in Matchmaker Gold Yeast-two-hybrid system. The fragment of *NcVg1* to *NcVg4* were separately cloned into pGBKT7 vector as the bait plasmid, which was then transformed into yeast strain AH109 to examine self-activation and toxicity. Bait plasmids and the cDNA library were then co-transformed in the

AH109 strain, with positive clones selected on SD/-Trp-Leu-His-Ade medium plates containing X-α-Gal (20 μg/mL; INALCO SPA Milano ITALY, 1758-0200) to assess β-galactosidase activity. The interaction between pGBKT7-53 and pGADT7-T served as the positive control, while the interaction between pGBKT7-Lam and pGADT7-T served as the negative control.

The ORFs of *NcRab5* and *NcRab27a* were separately cloned into the pGADT7 vector as prey plasmids, followed by co-transformation with NcVg2 bait plasmids into the AH109 yeast strain. The activity of β-galactosidase was assessed on SD/-Trp-Leu-His-Ade medium plates containing X-α-Gal (20 μg/mL).

A GST pull-down assay was conducted in accordance with previous methods (Chen et al., 2021). Specifically, the fragment of *NcVg2* was cloned into the pGEX-3x vector to construct plasmids expressing GST-fusion protein as the bait (GST-NcVg2), while the complete ORF of *NcRab5* was cloned into the pEASY-Blunt E1 Expression Vector (Transgen Biotech, Beijing, China) to construct plasmids expressing His-fusion protein as preys (His-NcRab5). The recombinant proteins fused with GST tag, as well as GST, were then separately expressed in the *Escherichia coli* strain BL21. Lysates were incubated with Glutathione-Sepharose beads (Cytiva, Sweden) before being subjected to incubation with recombinant proteins fused with His tag. The eluates were then analyzed by western blot assay using GST-tag and His-tag antibodies.

### Immunofluorescence staining of target proteins
Female adult salivary glands were dissected, fixed in 4% (w:v) paraformaldehyde, and permeabilized with 0.4% Triton X-100[17]. The samples were then treated with NcVg-FITC, NcRab27a- or NcRab5-rhodamine, and actin dyes (phalloidin-Alexa Fluor 647 carboxylic acid) for immunostaining. Finally, immunostained samples were analyzed using a TCS SP5 laser confocal microscope (Leica Microsystems).

To detect the release of target proteins in rice plants, 15 female adults were starved for 2 h, then fed on a region of 1.5 cm in length by 0.3 cm in width of a single rice seedling in a small cage for 24 h. The feeding areas of the tested rice seedlings were embedded with O.C.T. Compound and cut with a Shandon Cryotome FSE (Thermo Fisher Scientific) into sections 15 μm in depth. These sections were then immunolabeled with NcVg- or RDV-FITC, NcRab27a- or NcRab5-rhodamine, and processed for immunofluorescence microscopy. As a control, rice that was not exposed to the leafhopper treatment was treated identically.

For the establishment of immunofluorescence microscopy parameters, digital images with dimensions of 1024×1024 pixels were captured using specific excitation/emission wavelengths for FITC (495/517 nm), rhodamine (551/573 nm), and Alexa Fluor 647 (652/658 nm). These images were acquired using a 63 oil-immersion objective. To ensure consistency, samples within the same experimental group were subjected to the same immunofluorescence microscopy parameters in order to standardize background levels and capture images in a single section.

### Effect of knocking down *NcRab5* or *NcRab27a* on NcVg release
The T7 RNA polymerase promoter was added to the forward and reverse primers for the *NcRab27a* or *NcRab5* fragment to amplify a region of approximately 300–400 bp. The dsRNAs targeting NcRab27a (dsNcRab27a) and NcRab5 (dsNcRab5) were synthesized in vitro using the T7 RiboMAX Express RNAi System.

Newly emerged female leafhoppers were microinjected with dsNcRab5, dsNcRab27a, or dsGFP at the thorax intersegment region using a Nanoject II Auto-Nanoliter Injector (Spring) with approximately 200 ng of dsRNA per insect. The microinjected leafhoppers were then transferred to healthy rice seedlings for recovery. At 4 days post-injection, western blot assays were performed on salivary glands from 30 dsRNA-treated leafhoppers to determine the RNAi efficiency.

primary antibodies for NcRab27a-, NcRab5-, and NcVg-specific IgG were used, whereas goat anti-rabbit IgG-peroxidase served as the secondary antibody (Sangon Biotech, cat. D110058). Band intensities of proteins analyzed by western blot assays were quantified using ImageJ software.

The RT-qPCR was performed using 2 ×RealStar Fast SYBR qPCR Mix (Genstar, cat. A303), with the EF-1α transcript of salivary glands of 30 *N. cincticeps* insects serving as the internal reference for normalizing gene expression levels. Relative gene expression levels were calculated using the $2^{-\Delta\Delta CT}$ method.

At 4 days post-injection, approximately 50 dsRNA-treated leafhoppers were starved for 2 h, then fed on 1 rice seedling at the 2-leaf stage (approximately 10 cm in height) for 24 h. These plant samples were then tested for the presence of target proteins using western blot assays.

### Levels of JA, SA, H$_2$O$_2$, and related metabolites induced by nonviruliferous leafhopper
Approximately 30 female adults were starved for 2 h and then fed on 1 rice seedlings at the 2-leaf stage (approximately 10 cm in height) for 12 h. The content of JA and SA in the tested rice seedlings was analyzed using a UPLC-XEVO TQ-S MS triple quadrupole mass spectrometer (Waters, Milford, MA, USA) equipped with an ACQUITY UPLC BEH C18 column (2.1 × 100 mm, 1.7 μm) thermostatted at 40 °C. These tests were conducted at the Horticultural Biology and Metabolomics division of Fujian Agriculture and Forestry University. This experiment was performed in triplicate and repeated three times for each treatment.

The H$_2$O$_2$ content of the tested rice seedlings was determined using the Hydrogen Peroxide Assay Kit (Beyotime, cat. S0038). Approximately 10 mg of leaf tissues were homogenized with 200 μL of lysis buffer on ice. After centrifugation of 12, 000 g at 4 °C, the supernatant was added with 100 uL detection reagent and then tested absorbance at 560 nm. The standard curve was established based on standard substances with a series of known H$_2$O$_2$ content and tested absorbance at 240 nm. The absorbance of the samples and standard substances was measured using the SPARK 10 M Microplate spectrophotometer (TECAN, Austria).

The POD activity of the tested rice plants was determined using the Peroxidase Assay Kit (Nanjing Jiancheng Bioengineering Institute, cat. A084-3-1). Approximately 10 mg of leaf tissues were homogenized with 200 μL of 0.1 M PBS (pH 7.2–7.4) on ice. After centrifugation of 3, 500 g for 10 min at 4 °C, the supernatant was collected and successively incubated with reagents in the kit according to the user manual. The final samples were tested for absorbance at 420 nm. The standard curve was established based on standard substances with a series of known POD activity and tested absorbance at 420 nm. The absorbance of the samples and standard substances was measured using the spectrophotometer.

We determined the CAT activity of tested rice plants using the Catalase (CAT) Assay Kit (Nanjing Jiancheng Bioengineering Institute, cat. A007-1-1). Approximately 10 mg of leaf tissues were homogenized with 200 μL of 0.1 M PBS (pH 7.2–7.4) on ice. After centrifugation of 2,500 g for 10 min at 4 °C, the supernatant was collected and successively incubated with reagents in the kit according to the user manual. The final samples were tested for absorbance at 405 nm. The standard curve was established based on standard substances with a series of known CAT activity and tested absorbance at 405 nm. The absorbance of the samples and standard substances was measured using the spectrophotometer.

The MDA contents of tested rice plants were determined using the Malondialdehyde (MDA) Assay Kit (Nanjing Jiancheng Bioengineering Institute, cat. A003-1-2). Approximately 20 mg of leaf tissues were homogenized with 180 μL of 0.1 M PBS (pH 7.2–7.4) on ice. The samples were successively incubated with reagents in the kit, vortexed, and boiled in a water bath at 95 °C. The supernatant was collected from

cooling sample by centrifugation at 3,500 g for 10 min and was tested for absorbance at 532 nm. The standard curve was established based on standard substances with a series of known MDA contents and tested absorbance at 532 nm. The absorbance of the samples and standard substances was measured using the spectrophotometer.

### Relative expression of genes related to JA, SA, and ethylene in rice plants

Thirty leafhoppers (nonviruliferous or viruliferous) were fed on 1 rice seedling at the 2-leaf stage (approximately 10 cm in height) for 12 h, and the feeding rice seedlings were tested by RT-qPCR assays. The *Ubiquitin-conjugating enzyme E2* (*OsUBC*) transcript of *O. sativa* served as the internal reference, and the relative gene expression levels were calculated using the $2^{-\Delta\Delta CT}$ method.

### Visualization of $H_2O_2$ location and accumulation

Five female adults were starved for 2 h and then fed on a region of 1.5 cm in length by 0.3 cm in width of 1 rice seedling at the 2-leaf stage in a small cage for 12 h. The feeding areas of the tested rice seedlings were immersed in a 1 mg/mL solution of DAB (Merck, cat. D12384) and vacuumized at 60 kPa for 10 min, followed by incubation in the dark at 65 °C for 4 h. The samples were washed with absolute ethanol and examined under a light microscope (Nikon, DS-Ri2). This experiment was performed in triplicate and repeated three times for each treatment.

The feeding areas of the tested rice seedlings were also immersed in a 10 μM H2DCFDA (Merck, D6883), a fluorescent dye precursor for $H_2O_2$, and vacuumized at 60 kPa for 10 min, followed by incubation for 1–2 h in the dark. Then the samples were washed with sterile water in the dark three times, for 10–30 min each time. Finally, the samples were examined under the TCS SP5 laser confocal microscope (Leica Microsystems).

### Effect of *NcVg* knockdown on the rice defense to insects

A T7 RNA polymerase promoter with the sequence 5′-ATTCTCTA-GAAGCTTAATACGACTCACTATAGGG-3′ was added to the forward and reverse primers for the *NcVg2* and *GFP* genes at the 5′ terminal to amplify a region of 945 and 400 bp. The resulting dsRNAs, targeting NcVg2 (dsNcVg) and GFP (dsGFP), were synthesized in vitro using the T7 RiboMAX Express RNAi System (Promega Biotech, cat. P1700).

At 3 days post-emergence, nonviruliferous female adults were microinjected with dsNcVg or dsGFP (approximately 200 ng per insect). At 4 days post-injection, approximately 30 of these insects were tested in RT-qPCR and western blot assays to confirm RNAi efficiency. The mRNA expression level of *NcVg* in salivary glands was determined by RT-PCR assays. At 4 days post-injection, approximately 50 of these leafhoppers were starved for 2 h and then allowed to feed on 1 rice seedling at the 2-leaf stage (approximately 10 cm in height) for 24 h. Salivary glands or plant samples were collected and tested for the presence of NcVg using western blots. NcVg-specific IgG served as primary antibodies in western blot assays, while goat anti-rabbit IgG-peroxidase served as the secondary antibody (Sangon Biotech, cat. D110058). The band intensities of proteins analyzed by western blot assay were quantified using ImageJ software. These experiments were performed in triplicate and repeated three times for each treatment.

At 4 days post-injection, 1 rice seedling exposed to 30 of these female adults for 12 h were collected to determine the contents of JA, SA, $H_2O_2$, and MDA, as well as the activities of POD and CAT. The region of 1.5 cm in length of rice seedlings exposed to 5 female adults for 12 h were collected for DAB or H2DCFDA staining. These experiments were performed in triplicate and repeated three times for each treatment.

### Effect of *NcVg* knockdown on feeding behavior of leafhoppers

We conducted a GIGA-8 EPG system (Wageningen University, Wageningen, Netherlands)[68]. Rice seedlings were transplanted one day in advance into plastic pots filled with turf soil for the EPG assay. At 4 days

post microinjection of dsRNAs, each treated leafhopper was starved for 2 h, then anesthetized with $CO_2$ and attached to a gold wire (D = 20 μm, H = 10–15 cm) using water-soluble silver glue at the dorsal thorax. The wired insect was connected to the EPG probe, which was connected to an amplifier, and placed on the stem of the rice. A copper wire (D = 2 mm, H = 10 cm), connected to another amplifier and serving as the plant electrode, was inserted vertically into the pot soil. The EPG signals were digitized by a converter (DI710-UL, Dataq, Akron, USA), and the data were captured using Stylet + +a software (Wageningen University, Wageningen, Netherlands). Each leafhopper was continuously recorded for 3 h, and we recorded 13 valid biologically independent replicates. The EPG recordings were conducted in a quiet room at 25–28 °C, RH 70 ± 5%.

### Effect of RDV infection on NcVg release into rice plants

Second-instar nonviruliferous nymphs were allowed to feed on diseased rice plants for 8 days. At 14 days post-first access to diseased plants (padp), salivary glands from 30 viruliferous or nonviruliferous leafhoppers were dissected and analyzed for *NcVg* gene and protein expression using RT-qPCR and western blot assays.

To examine the release of target proteins into rice plants, 30 viruliferous or nonviruliferous leafhoppers were allowed to feed on 1 rice seedling at the 2-leaf stage for 2 days. The tested plants were then collected for protein extraction in equal quantities and the release of NcVg and RDV, as well as OsGSTF12 expression, were examined. Equal amounts of proteins from each treatment were loaded for western blot assays. This experiment was performed in triplicate and repeated three times for each treatment. Tested rice plants were also collected to examine activity of GST and contents of GSH and GSSG. Rice plants were also collected to examine GST activity and the contents of GSH and GSSG.

### Effect of GW4869 treatment on NcVg release

Newly emerged female leafhoppers were microinjected with GW4869 at a final concentration of 15 μg/ml. DMSO treatment served as a control. The microinjected leafhoppers were then transferred to healthy rice seedlings for recovery. At 4 days post-injection, western blot assays were performed on salivary glands from 30 GW4869-treated leafhoppers. Additionally, approximately 50 GW4869- or DMSO-treated leafhoppers were starved for 2 h, then fed on 1 rice seedling at the 2-leaf stage (approximately 10 cm in height) for 24 h. These plant samples were then tested for the presence of target proteins using western blot assays.

### Generation and insect resistance of NcVg-OE plants

To generate NcVg-OE plants, the DNA fragment of NcVg2 was first cloned into the pBWA(V)HS vector to form constructs 35 S:NcVg2. Then the constructs 35 S:NcVg2 was transformed *O. sativa* L.ssp. *Japonica* cv. Nipponbare using Agrobacterium tumefaciens-mediated transformation at Biorun Inc. (Wuhan, China). The $T_0$ transgenic lines that expressed NcVg2 were chosen and propagated for $T_1$ generation. Then the T1 transgenic lines that stably maintained the transgenes were determined by western blot assays and selected for phenotype analyses, leafhopper feeding, and viral infection assays.

One leaf of NcVg2-OE plants and WT plants at the tillering stage not exposed or exposed to 30 leafhoppers were analyzed for the activity of POD and CAT, as well as the contents of $H_2O_2$ and MDA using corresponding kits. To visualize $H_2O_2$ location and accumulation in NcVg2-OE plants, approximately 5 leafhoppers were allowed to feed on a region 1.5 cm in length by 0.5 cm in width of 1 leaf of NcVg2-OE plants and WT plants at the booting stage for 12 h. The feeding areas of the tested rice seedlings were treated with DAB or H2DCFDA, and samples were subsequently examined under a light microscope (Nikon, DS-Ri2) or TCS SP5 laser confocal microscope. These experiments were done in triplicate and repeated three times for each treatment.

To investigate the effect of NcVg2-OE plants on the plant penetration behavior of leafhoppers, newly emerged female adults were tested using the EPG technique. Transgenic lines or WT at the booting stage were used for the tests. Each insect was continuously recorded for 3 h, and at least 13 valid biologically independent replicates were recorded.

## H$_2$O$_2$ level and related metabolites in rice induced by viruliferous leafhopper

Nonviruliferous second-instar nymphs fed on diseased rice plants for 8 days to acquire RDV. Approximately 80% of the adults were confirmed as viruliferous. About 30 viruliferous or nonviruliferous adults were starved for 2 h and then fed on 1 rice seedlings at the 2-leaf stage (approximately 10 cm in height) for 12 h.

The tested rice seedlings were analyzed for the contents of JA and SA, as well as transcript levels of JA or SA related genes via RT-qPCR assays. These experiments were performed in triplicate and repeated three times for each treatment. The tested rice seedlings were also examined for contents of H$_2$O$_2$, MDA, GSH, and GSSG, as well as the activity of GST, POD, and CAT using the respective kits or through treatment with DAB or H2DCFDA, as mentioned above.

To evaluate the effect of RDV infection on the plant penetration behavior of leafhoppers, viruliferous or nonviruliferous adults were subjected to the EPG assay. Each insect was continuously recorded for 3 h, and at least 13 valid biologically independent replicates were recorded.

## Effect of *NcVg* knockdown of viruliferous leafhoppers on rice defense and viral transmission

Nonviruliferous second-instar nymphs were fed on diseased rice plants for 3 days. At 9 days post-padp, when the virus initially infected salivary glands, approximately 100 viruliferous or nonviruliferous leafhoppers were microinjected with dsNcVg or dsGFP (about 200 ng/insect).

At 14 days padp, salivary glands of 30 dsGFP- or dsNcVg-treated viruliferous leafhoppers were dissected and analyzed for the expression of NcVg and P8 by RT-qPCR and western blot assays. Thirty dsGFP- or dsNcVg-treated viruliferous leafhoppers were then fed on 1 rice seedling for 2 days. The tested plants were analyzed using western blot assays. This experiment was performed in triplicate and repeated three times for each treatment.

At 14 days padp, 30 dsRNA-treated viruliferous leafhoppers were allowed to feed on 1 rice seedling at the 2-leaf stage (approximately 10 cm in height) for 12 h to determine the contents of rice H$_2$O$_2$, MDA, GSH, and GSSG, as well as the activities of GST, POD, and CAT using the corresponding kits mentioned above. H$_2$O$_2$ localization and accumulation were tested using DAB or H2DCFDA. These experiments were performed in triplicate and repeated three times for each treatment.

At 14 days padp, the plant penetration behavior of dsRNA-treated viruliferous leafhoppers was analyzed using the EPG technique. Each insect was continuously recorded for three hours, and 13 valid biologically independent replicates were recorded.

To investigate the effect of *NcVg* knockdown on the profile of RDV transmission by leafhoppers, approximate 50 dsRNA-treated viruliferous leafhoppers individually inoculated a healthy rice seedling in one glass tube for two days at 14 days padp. All tested rice seedlings were planted in an insect-proof greenhouse for 60 days. The presence or absence of the RDV P8 gene was determined by RT-PCR assays.

## Interaction of NcVg with OsGSTF12

To screen the putative interactors of NcVg2 in rice plant used, a cDNA library of *O. sativa* L.ssp. *Japonica* cv. Nipponbare was constructed in the pGADT7 vector for prey plasmids in yeast two-hybrid system. The DNA fragment of NcVg2 was cloned into the pGBKT7 vector as the bait plasmids. The bait plasmids and the cDNA library were then co-transformed into AH109, and the transformants were subsequently

screened on the SD/-Trp-Leu, SD/-Trp-Leu-His, and SD/-Trp-Leu-His-Ade culture medium. Positive clones were selected on SD/-Ade-His-Leu-Trp plates containing X-α-Gal (20 µg/mL) to detect β-galactosidase activity. The interaction of pGBKT7-53 with pGADT7-T served as a positive control, while the interaction of pGBKT7-Lam with pGADT7-T served as a negative control.

To examine the NcVg-OsGSTF12 interaction in yeast two-hybrid system, the full-length ORF of *OsGSTF12* was cloned into the pGADT7 vector as prey plasmids. The prey and bait plasmids were co-transformed into AH109, and β-galactosidase activity was detected on SD/-Ade-His-Leu-Trp/X-a-Gal culture medium.

In BiFC assays, PCR-amplified DNA fragments of *NcVg2* and *OsGSTF12* were individually cloned into BiFC vectors YC and YN, respectively, to generate NcVg2-YC and OsGSTF12-YN. The resulting plasmids were transformed into *A. tumefaciens* strain GV3101, which was then infiltrated into leaf tissues of 4-week-old *N. benthamiana* plants. At 24–72 h post infiltration, the plant tissue samples were observed using a Leica TCS SP5 inverted confocal microscope. The interaction of NcVg2-YC with YN served as a negative control.

To examine the interaction of NcVg and OsGSTF12 in rice plants, approximately 30 leafhoppers were allowed to feed on a region of 1.5 cm in length by 0.3 cm in width of 1 leaf of rice seedling in a small cage for 24 h. The feeding areas of the tested rice seedlings were then embedded with O.C.T. Compound (Sakura, cat. 4583) and then sectioned. The sections were immunolabeled with NcVg-FITC, OsGSTF12-rhodamine, and processed for immunofluorescence microscopy. Rice plants not exposed to leafhoppers were treated the same way.

## Transient expression of OsGSTF12, NcVg 1 to NcVg4 in *N. benthamiana*

The fragments of *NcVg1* to *NcVg4* and full length ORF of *OsGSTF12* were separately cloned into the pCambia3301 vector to construct plasmids expressing GFP fusion protein. The plasmids were then transformed into *A. tumefaciens* strain GV3101, which was infiltrated into the leaf tissues of 4-week-old *N. benthamiana* plants. Samples of the plant tissues were observed using Leica TCS SP5 inverted confocal microscope or examined for the content of H$_2$O$_2$, GSH, and GSSG, as well as the activity of GST at 36–48 h post infiltration.

## GST activity, GSH, and GSSG contents

Approximately 30 female adult leafhoppers (nonviruliferous, viruliferous) were starved for 2 h and then fed on a region of 10 cm in length of a leaf of WT or NcVg2-OE plants for 12 h. The feeding areas were then tested for GST activity, and the contents of GSH and GSSG were determined. The activity of GST was calculated using a Glutathione S-Transferase (GST) Activity Assay Kit (Boxbio, cat. AKPR013M), while the content of GSH was determined using a Reduced Glutathione (GSH) Content Assay Kit (Boxbio, cat. AKPR008M), and the content of GSSG was determined using an Oxidized Glutathione (GSSG) Content Assay Kit (Boxbio, cat. AKPR009M). These experiments were performed in triplicate and repeated three times for each treatment.

To determine the effect of *NcVg* knockdown on OsGSTF12 suppressing H$_2$O$_2$, dsNcVg- or dsGFP-treated leafhoppers at 4 days post-microinjection were allowed to feed on rice plants for 12 h. The tested rice plants were then collected to examine OsGSTF12 expression, GST activity, and the contents of GSH and GSSG.

The reaction mixture was incubated with reagents in the kit following the user manual. Absorbance of the supernatant was measured at 412 nm, and the activity of GST (U/mg protein), content of GSH (µg/mg protein), and GSSG (µg/mg protein) of the sample were determined per kit instructions. These experiments were performed in triplicate and repeated three times for each treatment.

To examine the effect of *NcVg* knockdown on the suppression of H$_2$O$_2$ by OsGSTF12, approximately 80 dsNcVg- or dsGFP-treated

leafhoppers at 4 days post-microinjection were allowed to feed on rice plants for 12 h. The tested rice plants were then collected to examine OsGSTF12 expression, GST activity, and contents of GSH and GSSG.

### Generation and insect resistance of OsGSTF12-KO plants

OsGSTF12-KO plants were created by Biorun Bio-technology (Wuhan, China). The CRISPR/Cas9 constructs were introduced into *A. tumefaciens* strain EHA105 after being verified by sequencing, and then separately transferred into *O. sativa* L.ssp. *Japonica* cv. Nipponbare by Agrobacterium-mediated transformation. Homozygous $T_0$ plants, which were verified by sequencing, were self-pollinated to generate steady homozygous $T_1$ lines for further experiments. At the tillering stage, leaves of OsGSTF12-KO and WT plants were tested for activity of POD and CAT, contents of $H_2O_2$, MDA, GSH, and GSSG. Approximately 30 leafhoppers were starved for 2 h, and then were allowed to feed on a region of 1.5 cm in length by 0.5 cm in width of one leaf of OsGSTF12-KO and WT plants at tillering for 12 h. Then, the tested leaves were examined for activity of POD and CAT, contents of $H_2O_2$, MDA, GSH, and GSSG using the corresponding kits or treated with DAB or H2DCFDA, as mentioned above.

To examine the effect of OsGSTF12-KO plants on the plant penetration behavior of leafhoppers, OsGSTF12-KO plants and WT plants at booting stage were selected for the EPG assay. Each insect was continuously recorded for 3 h, and at least 13 valid biologically independent replicates were recorded.

### Exposure of NcVg2-OE or OsGSTF12-KO plants to viruliferous leafhoppers

Nonviruliferous second-instar nymphs fed on diseased rice plants for 8 days to acquire RDV. Approximately 50 viruliferous newly emerged female adults were starved for 2 h, then allowed to feed on 1 leaf of NcVg2-OE, OsGSTF12-KO, or WT plants at the tillering stage for 12 h. The tested leaves were examined for contents of $H_2O_2$, MDA, GSH, and GSSG, as well as the activity of GST, POD, and CAT, or treated with DAB or H2DCFDA, as mentioned above.

To test the transmission rate, approximate 50 viruliferous newly emerged female adults individually inoculated a healthy rice seedling in one glass tube for 2 days. All of the tested rice seedlings were planted in an insect-proof greenhouse for 60 days. RT-PCR was performed to test the presence or absence of the RDV P8 gene.

### Effect of prokaryotically expressed NcVg on $H_2O_2$ level and GST activity of rice plants

The fragments of *NcVg1* to *NcVg4* and full length of *GFP* were separately cloned into the pEASY-Blunt E1 Expression Vector to construct plasmids expressing fusion protein His-GFP, His-NcVg1, His-NcVg2, His-NcVg3 and His-NcVg4. The recombinant proteins were then separately expressed in the *E. coli* strain BL21.

Approximately 200 μL of lysates expressing His-GFP, His-NcVg1, His-NcVg2, His-NcVg3 or His-NcVg4 were injected into the stem base of rice seedlings at the two-leaf stage (approximately 10 cm in height). At 1-h post injection, the tested rice seedlings were examined for the contents of $H_2O_2$, GSH, and GSSG, as well as the activity of GST. This analysis was conducted in 3 independent biological replicates to ensure reliable results.

### Statistical analyses

All quantitative data presented in the figures were analyzed using two-tailed *t*-tests in GraphPad Prism 7 software (GraphPad Software, San Diego, CA, USA).

### Reporting summary

Further information on research design is available in the Nature Portfolio Reporting Summary linked to this article.

## Data availability

All data are available in the main text or the supplementary materials. Source data are provided with this paper.

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

## Acknowledgements

We are grateful to Dr. Toshihiro Omura (National Agricultural Research Center, Tsukuba, Japan) for providing the antibodies against intact viral particles. This work was supported by grants from the National Natural Science Foundation of China (Grants U21A20221) and the Natural Science Foundation of Fujian Province of China (Grants 2021J02010).

## Author contributions

T.W. and Q.C. designed the research; Y.W., C.L., S.G. and Y.G. conducted the experiments; Y.W. and C.L. contributed equally to this work; all authors analyzed the data; Q.C. wrote the paper. All authors read and approved the manuscript.

## Competing interests

Authors declare no competing interests.
