## [Peer Review File · Nature Communications]

Leafhopper salivary vitellogenin mediates virus transmission to plant phloemReviewer #1 (Remarks to the Author):

This manuscript investigated the reovirus Rice dwarf virus (RDV) transmission by the leafhopper *Nephotettix cincticeps*. The manuscript overall conclusion is that the virus uses salivary exosomes for release into the plant phloem. This exosome release involves also packaging the the insect vitellogenin through interaction with an insect GTPase Rab5 protein. The release into the phloem suppresses H₂O₂ burst by interaction with the rice plant GSTF12, thus facilitating insect feeding and subsequent virus transmission. this topic in arthropod – plant interactions and the role of effectors in modulating plant responses is intriguing, and there is an accumulating evidence of some effectors that are expressed from the insect genome to be secreted into plants and subsequently interact with plant proteins for modulating the plant immune response, thus enabling better feeding by the insect. This paper adds another trophic level which is the virus. The virus in this case mediate the transmission of the insect effector, in this case vitellogenin, to the plant phloem, where it interacts with plant proteins for modulating plant immunity. Some of the results need further calcifications and some of the steps of this research where not explained. I details here some major and minor comments for considerations:

1. Rab5 was identified following a Y2H screen. Where there other proteins identified in this screen? Why this protein and not others was chosen for further research. I think it is important to bring up the other results and discuss them as well.
 2. In figure 1E, a bright field should be provided and the cellular structures should be shown. I don't really know what is presented in this image and how this is relative to the whole cell, the nuclei and the rest of the cell. A zoom out might be necessary here.
 3. Figure 1G shows localization in the phloem, but again fails in good orientation. A bright field is shown, but the localization seems to be only in the periphery of the sieve element. Why?
 4. Again in figure C-I a bright field of the gland should be shown for good orientation.
 5. In EPG experiments the authors concluded that knockdown of Vg prolonged the duration of non-probing and shortened the duration of ingestion from phloem and xylem and had no effect on stylet penetration and salivations. The later two conclusions is not logical as H₂O₂ burst would definitely affect those two parameters and present a barrier for stylet penetrations. Some clarifications are needed.
 6. Where in the observations in figure 2D quantified and significant? Statistics should be shown.
 7. In all subsequent experiments i have no comments, the experiments were nicely conducted and supported by the presented results. The manuscript is well-packed and presented.
- I found two minor corrections that should be made. The paper should be read again for typos.
Line 54: no need for: which
Line 61: combined not combine

Reviewer #2 (Remarks to the Author):

The authors test the role of leafhopper vitellogenin (Vg) as a effector to facilitate RDV transmission.

First the authors localize by Western blot the uncleaved 220 kD Vg in salivary glands (SG) and in feeding sites on rice leaves. The authors do not indicate here which Vg protein they detect. Then they establish by yeast two hybrid and pulldown assays that Vg2 interacts with Rab5 GTPase (involved in exosome pathway). Immunofluorescence shows partial colocalization of Rab5 with Vg in vesicles in salivary glands. Rab5 is also found in feeding sites by western blot and immunofluorescence again indicates partial colocalization of Rab5 and Vg in phloem feeding sites. How do you explain the only partial colocalization in SG and feeding sites? Why is the Vg/Rab5 label in form of aggregates in figure 1G and diffuse in figure S1A? Then they show by siRNA that knockdown of Rab5 decreases Vg levels in salivary glands and feeding sites. As expected, also silencing Vg reduces Vg in SG and feeding sites. Then the authors claim that Vg does not seem to be released into plants via exosomes (in yeast double hybrid and immunofluorescence assays Vg does not seem to interact with Rab27a). However, there is partial colocalization of the exosome marker Rab27a and Vg in SG (figure S1C), so Rab27a could at least indirectly be involved in Vg translocation.

Then the authors show that feeding induces ROS in plants, silencing Vg increases ROS (figure 1K) indicating that Vg is an effector inhibiting ROS (and also decreasing phloem ingestion).

In the next series of experiments, RDV infection increases Vg, Rab5 and Rab27a (figure 2). RDV localizes with Vg in SG. Infection increases the number of vesicles in which Vg colocalizes with Rab5 or Rab27. This is not astonishing because there is more protein detected in infected insects. It would be more interesting to know whether infection increases the percentage of Vg containing vesicles. I believe this could easily be established using the existing data.

Knockdown of Rab27a and exosome inhibitor GW4869 decrease RDV, Rab27a and Vg in SG and plants, indicating a role for exosomes in their transport to the phloem. So contrary to the authors' claim, not only Rab5 but also Rab27a seems to play a role in Vg translocation.

In the next experiment, stable overexpression of Vg in healthy rice plants decreases ROS and improves feeding. On the other hand, RDV infection of insects increases ROS and punctures. This seems to be contradictory to RDV infection increasing Vg in vectors and rice (figure 2) and the positive effect of Vg on feeding, and could indicate that other factors play a role as well in vector-plant interactions.

Knockdown of Vg in infected insects decreases viral P8 titer in SG and feeding sites, and increases ROS. It also decreases transmission. This correlates with shorter phloem feeding, but longer salivation. This might support the idea that transmission-competent RDV is released during salivation in the phloem and not during salivation in the pathway phase. Taken together, as I understand it, there seem to be opposing effects of infection (worse feeding, more ROS) and Vg (better feeding, less ROS).

Next a mechanism is looked for to explain ROS inhibitory action of Vg. Yeast two hybrid and BIFC identified rice GSTF12 as interacting with Vg. Immunofluorescence shows that Vg and GSTF12 accumulate at feeding sites. In transient expression assays, GSTF12-GFP accumulates in nuclei, but this is not commented. GSTF12 is active in ROS scavenging (figure 6F). Insect feeding induces GSTF12 accumulation and activity. Comparison of feeding by infected or healthy insects shows that infected insects seem to stimulate higher GSTF12 accumulation and activity. At the same time, infected insects induce more ROS in feeding sites than do healthy insects. So there seems to be a discrepancy that the authors should clarify. Knocking down Vg in healthy or infected leafhoppers decreases GSTF12 accumulation and activity. The levels of GSH and GSSG do not always follow the same trend as GSTF12 accumulation (compare figure 6H with other GSH/GSSG histograms), which might indicate that also other enzymes control ROS status.

Next, GSTF12 knockout plants are created with reduced GST activity. They accumulate more ROS than WT plants and insect feeding is impaired on them.

Next, Vg overexpression and GSTF12 knockout plants are compared. A proviral effect of Vg overexpression and an antiviral effect of GSTF12 knockout is observed, similar to the results presented before in this paper.

Finally, transient expression of different domains of Vg shows that basically domain 2 is involved in ROS suppression.

The results are summarized in figure 9 as model. While the model is conclusive, I am missing integration of Rab27a in it. The authors should clarify this.

Taken together, this is an interesting manuscript and advances the field. My major concern is that there is some imprecision in the interpretation of data, for example the antagonism ROS/GST, or the role of Rab5 and Rab27a, respectively.

Other comments

Quantitative western blots: It is not clear to me whether the ratio values are means from the three replicates or the ratio from the blot analyzed in the corresponding figure.

EPG experiments: The added times of each behavior do not always match the total observation time (10800 s): figure 3D ~8600 s WT, ~10000 s Vg; figure 4F ~8000 s V-, ~7000 s V+; figure 5E ~7000 s GFP, 10000 s Vg; figure 7D ~8500 s WT, ~8500 s KO. Please explain.

Microscope settings and acquisition conditions should be described more in detail (excitation/emission wavelengths, stacks or single sections and so on).

Yeast two hybrid: The cDNA library should be described more in detail. Did you use fragmented

RNAs or full length mRNAs for library construction?

Line 129: Here is a discrepancy: Vg is released independent from exosome (because independent of Rab27a), line 181 it is exosome pathway because dependent of Rab5. Please clarify.

Line 193: why do you speak of complexes? They might just as well only colocalize in exosomes.

Line 226: You claim here a higher number of punctures, but figure 4E shows the opposite. Please clarify.

Reviewer #3 (Remarks to the Author):

Please see my comments below:

The function of Vg from small brown planthopper weakening plant defense by immunity regulator OsWRKY71 is known. Therefore, the role of Vg in modulating plant defense by Leafhopper Vg is not novel. This study now show the role of exosome-mediated transmission of Vg.

Authors need to provide strong justification for the significance of non-exosomal NcVg delivery during Nc feeding and exosomal NcVg delivery during Nc feeding and RDV transmission. They need to show why RDV prefers to have exosomal NcVg rather than salivary-secreted NcVg.

Lines 21-26-can be combined as one sentence. Currently, it reads redundant except feeding and viral transmission in two sentences.

Line 56- Expand genus species at the start of the sentence. *N. cincticeps*

Full blots/uncropped images for all immunoblots shown in the manuscript is required. Can be supplied as supplementary.

Reasoning for using NsVg1 and NsVg4 as bait proteins for Yeast-two hybrid analysis need justification.

It is unclear whether they have used NsVg2 or NsVg1 and NsVg4 for Y2H analysis. Line 106-110.

Again, Justification for use of NsVg2 is needed.

Effect of NcRab5 knockdown affects NcVg levels. The authors need to show what happens when NcVg is knockdown to NcRab5 levels and localization. Is there a feedback mechanism for association of both proteins?

Lines 126-130-lack of co-localization/interactions does not exclude that NcVg is secreted via exosomal independent pathway. Authors need to isolate exosomes and perform immunoblotting with both Rab27a and NcVg to confirm the absence of NcVb in exosomes.

Line 152 and Line 159- modify. It reads as if dsRNAs were extracted from NcVg knockdown leafhoppers. As I understood, rice seedlings were exposed to NcVg-dsRNA-treated Nc.

Line 174 and throughout-Authors need to clarify that rice seedlings are not treated with NcVg dsRNA but rather fed with NcVg-dsRNA-treated Nc.

Line 190-191-Modify. It reads as if NcVg antibody is cross-reactive with RDV.

Line 194-195-the authors need to quantify exosomes upon RDV infection to conclude this. There are several techniques available to quantify exosomes.

Line 206-Authors need to give some details on Nc-Vg transgenic plants. Construction methods..insertion site etc... Do NcVg-transgenic plants have any measurable phenotype compared

to the wild-type control?

Line 227 and Line 242-Authors need to explain why RDV-infected leafhoppers took long time for salivation and ingestion compared to the uninfected controls. If NcVg is good for leafhopper, should one see opposite if infected RDV leafhoppers produce more of NcVg right?

Authors need to explain why the observation of long time for salivation was noted in both wild-type RDV-infected leafhoppers and in NcVg-dsRNA treated-leafhoppers. The authors need to address this observation.

Lines 248-Authors should provide some evidence in cell lines or some sort of in vitro work showing effect of different doses of NcVg and OsGST12 on H₂O₂ production to claim that NcVg is responsible for inhibiting production of H₂O₂.

Line 253-Authors need to show what happens to OsGST12 levels upon RDV-infected leafhopper feeding compared to the levels noted upon feeding of non-infected controls.

Authors need to provide evidence whether NcVg directly induces expression of OsGST12 expression.

Lines 290- Authors need to give some details on how they generated OsGST12 knockout. Do OsGST12 has any measurable phenotype compared to wild-type control. It looks like roots are tiny in OsGST12 knockouts...This suggests that OsGST12 may also have additional functions.

Line 378-383-To claim that this is first report showing role of exosomes in RDV transmission, Following experiments must be addressed:

- a) Authors need to provide quantification on exosome numbers from non-infected and RDV-infected leafhoppers to claim that exosomes are induced upon rDV infection.
- b) Evidence to show that RDV is inside but not outside exosomes is needed.
- c) Evidence for absence of RDV in saliva is needed to claim that it is only present in exosomes.
- d) Quantification of NcVg inside and outside exosomes is needed to claim that RDV-infected leafhoppers secrete most or all of NcVg in exosomes but not in saliva.

Reviewer #1 (Remarks to the Author):

This manuscript investigated the reovirus Rice dwarf virus (RDV) transmission by the leafhopper *Nephotettix cincticeps*. The manuscript overall conclusion is that the virus uses salivary exosomes for release into the plant phloem. This exosome release involves also packaging the insect vitellogenin through interaction with an insect GTPase Rab5 protein. The release into the phloem suppresses H₂O₂ burst by interaction with the rice plant GSTF12, thus facilitating insect feeding and subsequent virus transmission. This topic in arthropod – plant interactions and the role of effectors in modulating plant responses is intriguing, and there is an accumulating evidence of some effectors that are expressed from the insect genome to be secreted into plants and subsequently interact with plant proteins for modulating the plant immune response, thus enabling better feeding by the insect. This paper adds another trophic level which is the virus. The virus in this case mediate the transmission of the insect effector, in this case vitellogenin, to the plant phloem, where it interacts with plant proteins for modulating plant immunity. Some of the results need further calcifications and some of the steps of this research where not explained. I details here some major and minor comments for considerations:

1. Rab5 was identified following a Y2H screen. Where there other proteins identified in this screen? Why this protein and not others was chosen for further research. I think it is important to bring up the other results and discuss them as well.

Response: Other protein identified in this screen was shown and analyzed in Table S1 and Fig. S1 of resubmitted version. Of putative interactors of NcVg2, 40 sequences were annotated using the BLASTX analyses in the GenBank. These putative interactors contained NcRab5, which regulates both intracellular and extracellular trafficking pathways and participates in RDV-packaging exosomes generation and delivery. Thus, Rab5 was chosen for further investigation.

We also discuss these putative interactors in the Discussion section: Putative interactors of NcVg2, such as vesicle-associated membrane protein, synaptobrevin, rabankyrin that is a Rab5 effector, in Y2H screening can be further studied, because they also associate with vesicle biogenesis and trafficking.

The related characterization was on lines 108-110, 418-420, Fig. S1 and supplementary Table 1.

Fig. S1 Analysis of putative interactors of NcVg2 in Y2H screening. a Species distribution percentage of putative interactors of NcVg2. Each pie presents the species hit by the putative interactors in BLASTX analyses. **b** Interaction frequency of more than 40 putative interactors of *N. cincticeps*. **c** GO categories of the putative interactors of *N. cincticeps* on molecular function. Nineteen functional groups of putative interactors with annotation were identified.

2. In figure 1E, a bright field should be provided and the cellular structures should be shown. I don't really know what is presented in this image and how this is relative to the whole cell, the nuclei and the rest of the cell. A zoom out might be necessary here.

Response: A bright field was added in resubmitted version. To demonstrate the whole cell, the nuclei and the rest of the cell, a schematic illustration of type III- to VI-cells in primary salivary gland from *N. cincticeps* was shown. A zoom out was also displayed. The revised Figure 1e was as follows.

Figure 1e Distribution of NcVg and NcRab5 in cavities and cytoplasm of salivary gland, as determined by immunofluorescence microscopy. The schematic illustration of type III- to VI-cells in salivary gland was shown. Panel i is the enlarged image of the boxed area. Panel showing green fluorescence (NcVg antigens), red fluorescence (NcRab5 antigens) or bright field is corresponding to panel i. APL, apical plasmalemma; Cv, cavity; N, nuclei; SD, salivary duct. Bars, 10 μm.

3. Figure 1G shows localization in the phloem, but again fails in good orientation. A bright field is shown, but the localization seems to be only in the periphery of the sieve element. Why?

Response: We are sorry to make you in puzzled. The whole cross section of the vein was added. The bundle sheath cell, epidermis cell, phloem, pitted vessel, sclerenchyma cell, sieve tubes were labeled. We hope these are helpful for the orientation.

Because the sieve tubes are conducting tissues, we believed that the interior periphery of sieve tubes are the main site of proteins accumulation.

Figure 1g Distribution of NcVg and NcRab5 in rice phloem, as determined by immunofluorescence microscopy. Panels ii is enlarged image of the boxed area. AS, air space; BSC, Bundle sheath cell; EC, Epidermis cell; P, phloem; PV, pitted vessel; SC, Sclerenchyma cell; ST, sieve tube. Bars, 10 μm.

4. Again in figure C-I a bright field of the gland should be shown for good orientation.

Response: The bright fields of salivary glands were added in figure 2c-i.

5. In EPG experiments the authors concluded that knockdown of Vg prolonged the duration of non-probing and shortened the duration of ingestion from phloem and xylem and had no effect on stylet penetration and salivations. The later two conclusions is not logical as H₂O₂ burst would definitely affect those two parameters and present a barrier for stylet penetrations. Some clarifications are needed.

Response: We are terribly sorry for this mistake. We checked and analyzed the data once again, and found the knockdown of NcVg prolonged the duration of salivation (NC3), but had no effect on stylet penetration (NC2). This suggests that dsNcVg-treated leafhoppers encountered barriers when feeding, and suppression of H₂O₂ burst require more effectors in saliva. It was also indicated that NcVg did not function in the process of stylet penetration. We revised the sentences on line 178-185.

6. Where in the observations in figure 2D quantified and significant? Statistics should be shown.

Response: We quantified the NcVg associated with vesicles in cavities. We found that the number of vesicles associated with NcVg antibody in cavities per section of one type III-cell of RDV-infected salivary glands was significantly higher than that of uninfected salivary glands. The related contents were shown in Fig. 2d-ii and on lines 205-207.

Fig. 2d-ii The mean number of vesicles associated with NcVg antibody in cavities per section of one type III-cell of one uninfected or infected salivary gland are shown. Fifteen random samples from infected or uninfected salivary glands were examined (two-tailed t-test).

7. In all subsequent experiments i have no comments, the experiments were nicely conducted and supported by the presented results. The manuscript is well-packed and presented.

I found two minor corrections that should be made. The paper should be read again for typos.

Response: We are sorry for these typos. We corrected them as possible as we could.

Line 54: no need for: which

Response: Thank you. We revised it.

Line 61: combined not combine

Response: Thank you. We revised it.

Reviewer #2 (Remarks to the Author):

The authors test the role of leafhopper vitellogenin (Vg) as a effector to facilitate RDV transmission.

First the authors localize by Western blot the uncleaved 220 kD Vg in salivary glands (SG) and in feeding sites on rice leaves. The authors do not indicate here which Vg protein they detect.

Response: The sentences related to type of vg protein was that “The 220-kDa NcVg was also detected in the salivary glands and rice seedlings exposed to leafhoppers (Fig.1b)”. These results indicate that intact NcVg can be secreted from salivary glands and released to the rice host.” They are on lines 100-103.

Then they establish by yeast two hybrid and pulldown assays that Vg2 interacts with Rab5 GTPase (involved in exosome pathway). Immunofluorescence shows partial colocalization of Rab5 with Vg in vesicles in salivary glands. Rab5 is also found in feeding sites by western blot and immunofluorescence again indicates partial colocalization of Rab5 and Vg in phloem feeding sites. How do you explain the only partial colocalization in SG and feeding sites?

Response: The possible reason of partial colocalization of NcRab5 and NcVg in SG is as follows. It is believed that NcRab5 also function in other processes independent of NcVg, including endosome trafficking. In the feeding sites, most NcRab5 colocalized with most NcVg. Very few NcVg antigens not colocalized with NcRab5, which is postulated that NcRab5 also functions in other mechanisms independent of NcVg. We added this explanation to Discussion section on lines 476-479.

Why is the Vg/Rab5 label in form of aggregates in figure 1G and diffuse in figure S1A?

Response: We are sorry for making you confused. Fig. S1a showed rice section not exposed to leafhopper, therefore, there is no specific signal of Vg or Rab5. These panels are background. When the rice plant was exposed to leafhoppers, Vg/Rab5 label in form of aggregates, as shown in Fig. 1g. We added “Nf” to signify “not exposed to leafhopper” in Fig. S2a of resubmitted version.

Then they show by siRNA that knockdown of Rab5 decreases Vg levels in salivary glands and feeding sites. As expected, also silencing Vg reduces Vg in SG and feeding sites.

Then the authors claim that Vg does not seem to be released into plants via exosomes (in yeast double hybrid and immunofluorescence assays Vg does not seem to interact with Rab27a). However, there is partial colocalization of the exosome marker Rab27a and Vg in SG (figure S1C), so Rab27a could at least indirectly be involved in Vg translocation.

Response: Yes, only a few NcVg antigens colocalizing with NcRab27a within the cytoplasm and salivary cavities (Fig. S2d). It was suggested that most NcVg antigens are released from salivary glands independent of the exosomal pathway in nonviruliferous leafhoppers, while only a few NcVg antigens are released depending on exosomal pathway.

Then the authors show that feeding induces ROS in plants, silencing Vg increases ROS (figure 1K) indicating that Vg is a effector inhibiting ROS (and also decreasing phloem ingestion).

In the next series of experiments, RDV infection increases Vg, Rab5 and Rab27a (figure 2). RDV localizes with Vg in SG. Infection increases the number of vesicles in which Vg colocalizes with Rab5 or Rab27. This is not astonishing because there is more protein detected in infected insects. It would be more interesting to know whether infection increases the percentage of Vg containing vesicles. I believe this could easily be established using the existing data.

Response: The quantified analyses were shown in Fig. 2c-ii and 2d-ii in resubmitted version. We found significantly increasing number of NcVg colocalized with NcRab5 or NcRab27a within infected salivary glands in immunofluorescence microscopy and increasing number of vesicles associated with NcVg antibody in cavities of infected salivary glands in immunoelectron microscopy. It was suggested that RDV infection likely caused salivary glands likely secrete more NcVg in exosomes than in saliva. The related contents were on lines 200-209.

Fig. 2c-ii The mean number of NcVg colocalizing with NcRab5 or NcRab27a in uninfected or infected salivary gland are shown in **c-ii**. Fifteen random 20x20 μm fields of samples from infected or uninfected salivary glands were examined in immunofluorescence microscopy.

Fig. 2d-ii The mean number of vesicles associated with NcVg antibody in cavities per section of one type III-cell of one uninfected or infected salivary gland are shown in **d-ii**. Fifteen random samples from infected or uninfected salivary glands were examined in immunoelectron microscopy.

Knockdown of Rab27a and exosome inhibitor GW4869 decrease RDV, Rab27a and Vg in SG and plants, indicating a role for exosomes in their transport to the phloem. So contrary to the authors' claim, not only Rab5 but also Rab27a seems to play a role in Vg translocation.

Response: Yes, Rab27a, serving as the marker of exosome, play a role in Vg translocation from RDV-infected salivary glands. Both knockdown of Rab27a or treatment of exosome inhibitor GW4869 in RDV-infected salivary glands proved the translocation function of exosome for Vg. Here, we underlined the function of exosome in the process of Vg translocation, but not exosomal components.

In the next experiment, stable overexpression of Vg in healthy rice plants decreases ROS and improves feeding. On the other hand, RDV infection of insects increases ROS and punctures. This seems to be contradictory to RDV infection increasing Vg in vectors and rice (figure 2) and the positive effect of Vg on feeding, and could indicate that other factors play a role as well in vector-plant interactions.

Response: We found that RDV infection in leafhoppers increased H₂O₂ burst of rice and made it more difficult for the leafhoppers to feed, indicating an enhancement in plant defense to insects. Viral infection of insects generally not only triggers the production of various salivary effectors like Vg, but also salivary elicitors, which trigger the defense of plant host to insects. Therefore, the increased defense of rice host caused by RDV-infected leafhopper is believed to be comprehensive. We demonstrated the results of RDV infection of leafhoppers to underline the rice defense to RDV-infected leafhoppers. The integrated effects of RDV infection on rice defense is not equivalent to individual effect of NcVg on rice defense. We don't think it is contradictory between RDV infection of insects increasing ROS and the the positive effect of NcVg on leafhopper feeding. We also added these explanations to the Discussion section on lines 383-396.

Knockdown of Vg in infected insects decreases viral P8 titer in SG and feeding sites, and increases ROS. It also decreases transmission. This correlates with shorter phloem feeding, but longer salivation. This might support the idea that transmission-competent RDV is released during salivation in the phloem and not during salivation in the pathway phase. Taken together, as I understand it, there seem to be opposing effects of infection (worse feeding, more ROS) and Vg (better feeding, less ROS).

Response: The core point of this comments is similar to last one. The rice defense to RDV-infected leafhoppers is caused by several salivary factors, including salivary elicitors, which induce the defense of plant host to insects. The integrated effects of RDV infection on rice defense is not equivalent to individual effect of NcVg on rice defense. We demonstrated the results of RDV infection of leafhoppers to underline the rice defense to RDV-infected leafhoppers. We also added these explanations to the Discussion section on lines 383-396.

Next a mechanism is looked for to explain ROS inhibitory action of Vg. Yeast two hybrid and BIFC identified rice GSTF12 as interacting with Vg. Immunofluorescence shows that Vg and GSTF12

accumulate at feeding sites. In transient expression assays, GSTF12-GFP accumulates in nuclei, but this is not commented.

Response: It was unknown whether the localization or accumulation of GSTF12-GFP in *N. benthamiana* was similar to that of GSTF12 in rice plants, therefore, we did not provide comments on GSTF12-GFP accumulates in nuclei. We demonstrated Fig. 6e was just to prove the expression of GSTF12-GFP in *N. benthamiana*.

GSTF12 is active in ROS scavenging (figure 6F). Insect feeding induces GSTF12 accumulation and activity. Comparison of feeding by infected or healthy insects shows that infected insects seem to stimulate higher GSTF12 accumulation and activity. At the same time, infected insects induce more ROS in feeding sites than do healthy insects. So there seems to be a discrepancy that the authors should clarify.

Response: The core point of this comments is similar to last two questions. Exposure of plant host to RDV-infected leafhoppers did cause high level of ROS, because RDV infection triggers the production of various salivary effectors and elicitors, which trigger plant defense. During this process, one such salivary protein NcVg upregulated by RDV induced more OsGSTF12 through interaction to suppress H₂O₂ burst. In other words, the OsGSTF12, acting as an interactor, responds to NcVg. Therefore, the effect of NcVg on rice defense was not equivalent to integrated effects of RDV infection on rice defense. Without the effectors like NcVg suppressing ROS burst or other plant defense, insects would be unable to feed on plant host. Thus, there is no discrepancy between higher ROS and higher OsGSTF12 caused by exposure to RDV infected leafhoppers. We also added these explanations to Discussion on lines 383-396.

Knocking down Vg in healthy or infected leafhoppers decreases GSTF12 accumulation and activity. The levels of GSH and GSSG do not always follow the same trend as GSTF12 accumulation (compare figure 6H with other GSH/GSSG histograms), which might indicate that also other enzymes control ROS status.

Response: Yes, other salivary factors also control ROS status.

Next, GSTF12 knockout plants are created with reduced GST activity. They accumulate more ROS than WT plants and insect feeding is impaired on them.

Next, Vg overexpression and GSTF12 knockout plants are compared. A proviral effect of Vg overexpression and an antiviral effect of GSTF12 knockout is observed, similar to the results presented before in this paper.

Finally, transient expression of different domains of Vg shows that basically domain 2 is involved in ROS suppression.

The results are summarized in figure 9 as model. While the model is conclusive, I am missing integration of Rab27a in it. The authors should clarify this.

Response: Thank you. We added Rab27a to the model in figure 9.

Taken together, this is an interesting manuscript and advances the field. My major concerns is that there is some imprecision in the interpretation of data, for example the antagonism ROS/GST, or the role of Rab5 and Rab27a, respectively.

Response: The GSH-dependent peroxidase activities of GSTs can scavenge toxic hydroperoxides to protect from ROS, oxidative damage and maintain cellular redox homeostasis (Dixon and Edwards, 2010; Moons, 2005). These GST enzymes use GSH as an electron donor to reduce hydroperoxides

and generate GSSG, the oxidized form of GSH (Moons, 2005). We added these contents to the Discussion section on lines 429-433.

The roles of Rab5 and Rab27a are as follows. Rabs are the key regulators of membrane identity and intercellular vesicle budding, trafficking and fusion, even exosome formation and trafficking (Anderson et al., 2016; Schorey et al., 2015). Several Rabs are implicated in the biogenesis and release of exosomes, including Rab27 and Rab5 (Akers et al., 2013; Alenquer and Amorim, 2015; Zeigerer et al., 2015). Rab5 generally localizes to early endosomes where they drive multiple aspects of endosome trafficking, and is also detected within the exosomes (Anderson et al., 2016). Rab27 regulates the fusion of late endosomes at the plasma membrane and functions in the docking site to release exosomes (Schorey et al., 2015). Many viruses exploit Rabs to their advantage, such as hepatitis C virus and Herpes simplex virus-1 (Salloum et al., 2013; Temme et al., 2010). We added these contents to the Discussion section on lines 454-461.

References:

- Dixon, D.P., Edwards, R. 2010. Glutathione transferases. *Arabidopsis Book* 8, e0131.
- Moons, A. 2005. Regulatory and functional interactions of plant growth regulators and plant glutathione S-transferases (GSTs). *Vitam Horm* 72, 155-202.
- Akers, J.C., Gonda, D., Kim, R., Carter, B.S., Chen, C.C., 2013. Biogenesis of extracellular vesicles (EV): exosomes, microvesicles, retrovirus-like vesicles, and apoptotic bodies. *J. Neurooncol.* 113, 1-11.
- Alenquer, M., Amorim, M.J., 2015. Exosome Biogenesis, Regulation, and Function in Viral Infection. *Viruses* 7, 5066-5083.
- Anderson, M.R., Kashanchi, F., Jacobson, S., 2016. Exosomes in Viral Disease. *Neurotherapeutics* 13, 535-546.
- Salloum, S., Wang, H., Ferguson, C., Parton, R.G., Tai, A.W., 2013. Rab18 binds to hepatitis C virus NS5A and promotes interaction between sites of viral replication and lipid droplets. *PLoS Pathog* 9, e1003513.
- Schorey, J.S., Cheng, Y., Singh, P.P., Smith, V.L., 2015. Exosomes and other extracellular vesicles in host-pathogen interactions. *EMBO Rep* 16, 24-43.
- Temme, S., Eis-Hübinger, A.M., McLellan, A.D., Koch, N., 2010. The herpes simplex virus-1 encoded glycoprotein B diverts LA-DR into the exosome pathway. *J. Immunol.* 184, 236-243.
- Zeigerer, A., Bogorad, R.L., Sharma, K., Gilleron, J., Seifert, S., Sales, S., Berndt, N., Bulik, S., Marsico, G., D'Souza, R.C.J., Lakshmanaperumal, N., Meganathan, K., Natarajan, K., Sachinidis, A., Dahl, A., Holzhutter, H.G., Shevchenko, A., Mann, M., Koteliansky, V., Zerial, M., 2015. Regulation of liver metabolism by the endosomal GTPase Rab5. *Cell Rep.* 11, 884-892.

Other comments

Quantitative western blots: It is not clear to me whether the ratio values are means from the three replicates or the ratio from the blot analyzed in the corresponding figure.

Response: The ratio values are from the blot analyzed in the corresponding figure.

EPG experiments: The added times of each behavior do not always match the total observation time (10800 s): figure 3D ~8600 s WT, ~10000 s Vg; figure 4F ~8000 s V-, ~7000 s V+; figure 5E ~7000 s GFP, 10000 s Vg; figure 7D ~8500 s WT, ~8500 s KO. Please explain.

Response: Yes, the total duration of observation for each leafhopper was 10800 seconds (3 hours). In this process, except for the feeding or stylet penetration, leafhopper also had other activities, like flying. Therefore, the duration of other activities was not detected in EPG assays.

Microscope settings and acquisition conditions should be described more in detail (excitation/emission wavelengths, stacks or single sections and so on).

For the establishment of immunofluorescence microscopy parameters, digital images with dimensions of 1024×1024 pixels were captured using specific excitation/emission wavelengths for FITC (495/517 nm), rhodamine (551/573 nm), and Alexa Fluor 647 (652/658 nm). These images were acquired using a 63 oil-immersion objective. To ensure consistency, samples within the same experimental group were subjected to the same immunofluorescence microscopy parameters in order to standardize background levels and capture images in a single section.

Yeast two hybrid: The cDNA library should be described more in detail. Did you use fragmented RNAs or full length mRNAs for library construction?

Response: We are sorry for the lack of cDNA library details. The total RNAs of *N. cincticeps* were isolated for a cDNA library construction by Oebiotech company of China. The cDNA library was constructed in pGADT7 vector (Clontech, K1612-1) for screening putative interactors of NcVg in Matchmaker Gold Yeast-two-hybrid system. The related contents were on lines 526-529.

Line 129: Here is a discrepancy: Vg is released independent from exosome (because independent of Rab27a), line 181 it is exosome pathway because dependent of Rab5. Please clarify.

Response: We are sorry for these unclear conclusions. We revised the sentences as “most NcVg antigens are released from salivary glands independent of exosomal pathway in nonviruliferous leafhoppers” on lines 137-138 and “NcVg is likely packaged in RDV-induced exosomes in salivary glands of viruliferous leafhoppers” on line 196.

Regarding the different modes of NcVg release from salivary glands of viruliferous and nonviruliferous leafhoppers, it was postulated that the proportion of exosomal NcVg was the key reason. Our previous study revealed a low abundance of exosomes in the cavities of uninfected salivary glands, whereas RDV infection induces a significant increase in the number of exosomes (Chen et al, eLife, 2021). In this study, only a small fraction of NcVg antigens were within exosomes in cavities of uninfected salivary glands. This suggests that the low proportion of exosomal NcVg is likely attributed to the limited number of exosomes. In contrast, in the RDV-infected salivary glands, the number of exosome increased and NcVg entered the virus-induced exosomes through the interaction between NcVg2 and NcRab5. Consequently, this led to a higher proportion of exosomal NcVg. Therefore, release of exosomal NcVg from RDV-infected salivary glands was NcRab5 dependent. We concluded that RDV hijacks NcVg to release from salivary glands via exosomal release pathway. There is no discrepancy between Vg is released independent from exosome in uninfected salivary glands and Vg is released dependent on exosomes in infected salivary glands. We also discussed this problem in Discussion section on lines 405-416.

Reference:

Chen et al. Exosomes mediate horizontal transmission of viral pathogens from insect vectors to plant phloem. *eLife* 2021;10:e64603

Line 193: why do you speak of complexes? They might just as well only colocalize in exosomes.

Response: Previous study revealed that RDV P2 protein interacts with NcRab5, and in this study we confirmed that NcVg2 interacted with NcRab5. Therefore, the P2-NcRab5-NcVg2 interaction, as well as colocalization of RDV, NcVg and NcRab5, indicated that RDV-NcVg-NcRab5 form complexes. We added these explanations to lines 192-194.

Line 226: You claim here a higher number of punctures, but figure 4E shows the opposite. Please clarify.

Response: We are terribly sorry for this mistake. We analyzed the number of punctures again, and showed the revised figure 4e in resubmitted version.

Reviewer #3 (Remarks to the Author):

Please see my comments below:

The function of Vg from small brown planthopper weakening plant defense by immunity regulator OsWRKY71 is known. Therefore, the role of Vg in modulating plant defense by Leafhopper Vg is not novel. This study now show the role of exosome-mediated transmission of Vg.

Authors need to provide strong justification for the significance of non-exosomal NcVg delivery during Nc feeding and exosomal NcVg delivery during Nc feeding and RDV transmission. They need to show why RDV prefers to have exosomal NcVg rather than salivary-secreted NcVg.

Response: Regarding the different modes of NcVg release from salivary glands of viruliferous and nonviruliferous leafhoppers, it was postulated that the proportion of exosomal NcVg was the key reason. Our previous study revealed a low abundance of exosomes in the cavities of uninfected salivary glands, whereas RDV infection induces a significant increase in the number of exosome (Chen et al, *eLife*, 2021). In this study, only a small fraction of NcVg antigens were within exosomes in cavities of uninfected salivary glands. This suggests that the low proportion of exosomal NcVg is likely attributed to the limited number of exosomes. Thereby, salivary NcVg in uninfected salivary glands was delivered in a non-exosomal manner.

In contrast, in RDV-infected salivary glands, the number of exosome increased and NcVg entered the virus-induced exosomes through the interaction between NcVg2 and NcRab5. Consequently, this led to a higher proportion of exosomal NcVg. Therefore, as we observed, the colocalization number of Vg and Rab27a was increased, and more NcVg antigens localized in virus-packaging exosomes. Thus, we would not conclude that RDV prefers exosomal NcVg, but RDV induces more exosomes, leading to increased proportion of exosomal NcVg. We also discussed this problem in Discussion section on lines 405-416.

Reference:

Chen et al. Exosomes mediate horizontal transmission of viral pathogens from insect vectors to plant phloem. *eLife* 2021;10:e64603

Lines 21-26-can be combined as one sentence. Currently, it reads redundant except feeding and viral transmission in two sentences.

Response: Thank you. We revised it.

Line 56- Expand genus species at the start of the sentence. *N. cincticeps*

Response: Thank you. We revised it.

Full blots/uncropped images for all immunoblots shown in the manuscript is required. Can be supplied as supplementary.

Response: Thank you. We showed the uncropped images for all immunoblots in supplementary file.

Reasoning for using NsVg1 and NsVg4 as bait proteins for Yeast-two hybrid analysis need justification.

Response: NcVg1 and NcVg2 respectively covers two different vitellogenin N domains, NcVg3 covers a domain of unknown function, NcVg4 covers a von Willebrand factor type D domain. Therefore, NsVg1 to NsVg4 were used as bait proteins. We added these explanations to lines 104-106.

It is unclear whether they have used NsVg2 or NsVg1 and NsVg4 for Y2H analysis. Line 106-110.

Response: We are sorry for making you confused. Yes, NsVg2, NsVg1 and NsVg4 were used for Y2H analysis. We change "NcVg1 to NcVg4" to "NcVg1, NcVg2, NcVg3 and NcVg4" in resubmitted version. Again, Justification for use of NsVg2 is needed.

Response: NcVg2 covers a vitellogenin N domain. Therefore, NsVg2 were used as bait proteins. We added these explanations to lines 104-106.

Effect of NcRab5 knockdown affects NcVg levels. The authors need to show what happens when NcVg is knockdown to NcRab5 levels and localization. Is there a feedback mechanism for association of both proteins?

Response: We found that knockdown of NcVg had no significant effect on transcription and expression of NcRab5. It was suggested the absence of feedback mechanism for association of NcVg and NcRab5. The related contents were on lines 127-130.

Lines 126-130-lack of co-localization/interactions does not exclude that NcVg is secreted via exosomal independent pathway. Authors need to isolate exosomes and perform immunoblotting with both Rab27a and NcVg to confirm the absence of NcVb in exosomes.

Response: Yes, co-localization and interactions could not completely support that NcVg is secreted via exosomal pathway. But we knocked down NcRab27a or used the GW4869, which is the specific exosome inhibitor, to block exosome biogenesis. We found the reduced accumulation of NcVg in salivary glands and resulting decreased release of NcVg to rice seedlings. Therefore, we concluded that NcVg is secreted via exosomal dependent pathway.

It seems impossible for us to isolate exosomes from salivary glands from leafhoppers, because the leafhoppers are so tiny that the exosome isolated from salivary glands using the general method are very limited.

Line 152 and Line 159- modify. It reads as if dsRNAs were extracted from NcVg knockdown leafhoppers. As I understood, rice seedlings were exposed to NcVg-dsRNA-treated Nc.

Response: We are terribly sorry for making puzzle. We changed “derived from” to “targeting”.

Line 174 and throughout-Authors need to clarify that rice seedlings are not treated with NcVg dsRNA but rather fed with NcVg-dsRNA-treated Nc.

Response: We are terribly sorry for making puzzle. We changed “dsNcVg treatment” to “exposure of rice plants to dsNcVg-treated leafhoppers”.

Line 190-191-Modify. It reads as if NcVg antibody is cross-reactive with RDV.

Response: We are terribly sorry for making puzzle. We changed “reacted with” to “colocalized with”.

Line 194-195-the authors need to quantify exosomes upon RDV infection to conclude this. There are several techniques available to quantify exosomes.

Response: In previous study, we quantified exosomes induce by RDV infection in salivary glands of leafhopper (Chen et al, 2021, eLife). We found that viral infection induces a significant increase of exosomes in salivary glands.

Reference:

Chen et al. Exosomes mediate horizontal transmission of viral pathogens from insect vectors to plant phloem. eLife 2021;10:e64603

Line 206-Authors need to give some details on Nc-Vg transgenic plants. Construction methods..insertion site etc ... Do NcVg-transgenic plants have any measurable phenotype compared to the wild-type control?

Response: The details of NcVg2 transgenic plants generation was as follows. To generate NcVg-OE plants, the DNA fragment of NcVg2 was first cloned into the pBWA(V)HS vector to form construct 35S:NcVg2. Then the construct 35S:NcVg2 was transformed *O. sativa* L.ssp. *Japonica* cv. Nipponbare using *Agrobacterium tumefaciens*-mediated transformation at Biorun Inc. (Wuhan, China). The T₀ transgenic lines that expressed NcVg2 were chosen and propagated for T₁ generation. Then the T₁ transgenic lines that stably maintained the transgenes were determined by western blot assays and selected for phenotype analyses, leafhopper feeding, and viral infection assays. This content was shown on lines 756-763.

We provided the measurable phenotype of NcVg-transgenic plants compared to the WT. We found that lines #3 and #12 exhibited a clear decrease in grain length and width, compared to WT (Fig. S5c and d). The related data were shown in Fig. S5

Fig. S5 Transient expression of NcVg1 to NcVg4 and characterization of basic properties of NcVg2-OE transgenic plants. **a** Comparison of phenotypic traits between WT and NcVg2-OE transgenic plants. **b** Overexpression of NcVg2 in NcVg2-OE transgenic plants from #1 to #14, as determined by western blot assays. The proteins were detected by using NcVg-specific antibody in western blot assays. Data represent 3 biological replicates. **c** Grain phenotypes of lines #1 and #5, as well as WT in the background of *O. sativa* L.ssp. *Japonica*, variety *Nipponbare*. Bars, 10 mm. **d** Measurement of grain length and width of the WT, #1 and #5 lines.

Line 227 and Line 242-Authors need to explain why RDV-infected leafhoppers took long time for salivation and ingestion compared to the uninfected controls. If NcVg is good for leafhopper, should one see opposite if infected RDV leafhoppers produce more of NcVg right?

Response: RDV infection of *N. cincticeps* took long time for salivation and ingestion, indicating the increased difficulty of viruliferous leafhoppers in feeding. It was suggested that the defense of plant host to insects was improved. Viral infection of insects generally not only triggers the production of various salivary effectors like Vg, but also insect elicitors, which trigger the plant defense. Therefore, the increased defense of rice host caused by RDV-infected leafhopper is believed to be comprehensive. We demonstrated the results of RDV infection of leafhoppers to underline the rice defense to RDV-infected leafhoppers. The integrated effects of RDV infection on rice defense is not equivalent to individual effect of NcVg on rice defense. There is no contradiction between feeding difficulty caused by RDV infection of leafhoppers and the positive effect of NcVg on leafhopper

feeding. We also added these explanations to Discussion on lines 383-396.

Authors need to explain why the observation of long time for salivation was noted in both wild-type RDV-infected leafhoppers and in NcVg-dsRNA treated-leafhoppers. The authors need to address this observation.

Response: The core point of this comments is similar to previous one and comments of reviewer #2. RDV infection of *N. cincticeps* took long time for salivation and ingestion, indicating the increased difficulty of viruliferous leafhoppers in feeding. It was suggested that the defense of plant host to insects was improved. When NcVg was knockdown, leafhoppers had to prolong duration of salivation to secrete more salivary effectors including NcVg at a low level to suppress H₂O₂ burst. Therefore, the individual effect of NcVg on rice defense was not equivalent to integrated effects of RDV infection on rice defense. We also added these explanations to Discussion on lines 383-396.

Lines 248-Authors should provide some evidence in cell lines or some sort of in vitro work showing effect of different doses of NcVg and OsGST12 on H₂O₂ production to claim that NcVg is responsible for inhibiting production of H₂O₂.

Response: We expressed NcVg1 to NcVg4 in *N. benthamiana*, and found that leaves transiently expressing NcVg2 had the lowest content of H₂O₂ and GSH, and the highest activity of GST, compared to those expressing NcVg1, NcVg3, or NcVg4 (Fig. 8f). In resubmitted version, we added the evidences from rice plants. We performed the injection of stem bases of rice seedlings with prokaryotically expressed NcVg1 to NcVg4 in equal doses. We found that the injection with NcVg2 caused significantly lower level of H₂O₂, and GSH in rice seedlings while higher activity of GST and content of GSSG, compared to injection with NcVg1, NcVg3, or NcVg4. Therefore, NcVg2 was responsible for inhibiting the production of H₂O₂. Related contents were shown on lines 364-369 and Fig. 8g.

Fig.8g Injection of rice seedlings with His-Vg2 significantly reducing contents of H₂O₂ and GSH accumulation, increasing GST activity and GSSG level of rice plant. Means (\pm SD) represent three biological replicates (two-tailed t-test). Ns, not significant.

Line 253-Authors need to show what happens to OsGST12 levels upon RDV-infected leafhopper feeding compared to the levels noted upon feeding of non-infected controls.

Response: The OsGST12 levels upon RDV-infected leafhopper feeding was shown in Figure 6i and j. We found that rice seedlings exposed to viruliferous leafhoppers had higher accumulation of OsGST12, activity of GST, and production of GSSG, while having lower content of GSH in rice plants, compared with rice seedlings exposed to nonviruliferous leafhoppers.

Authors need to provide evidence whether NcVg directly induces expression of OsGST12 expression.

Response: We performed the injection of stem bases of rice seedlings with prokaryotically expressed NcVg1 to NcVg4 in equal amounts. We found that the injection with NcVg2 caused significantly lower level of H₂O₂, and GSH in rice seedlings while higher activity of GST and content of GSSG, compared to injection with NcVg1, NcVg3, or NcVg4. Together with evidences of NcVg-OsGSTF12 interaction, and upregulated expression of OsGSTF12 and catalysis of GST in Vg2-overexpressing transgenic plant, we concluded that NcVg could directly induce expression and activity of OsGSTF12. Related contents were shown on lines 364-369 and Fig. 8g.

Fig.8g Injection of rice seedlings with His-Vg2 significantly reducing contents of H₂O₂ and GSH accumulation, increasing GST activity and GSSG level of rice plant. Means (± SD) represent three biological replicates (two-tailed t-test). Ns, not significant.

Lines 290- Authors need to give some details on how they generated OsGSTF12 knockout. Do OsGSTF12 has any measurable phenotype compared to wild-type control. It looks like roots are tiny in OsGSTF12 knockouts...This suggests that OsGSTF12 may also have additional functions.

Response: We used the CRISPR/Cas9 technology to generate OsGSTF12-knockout (KO) transgenic plants. Sequences of *OsGSTF12* gene of OsGSTF12-KO plants showed the substitution or deletion in different number of nucleic acids (Fig. S7a). These OsGSTF12-KO plants displayed varying activities of GST (Fig. S7c).

We provided the measurable phenotype of OsGSTF12-KO plants compared to the WT in resubmitted version. We found that Lines #12 and #29 displayed not significant change in grain length, but a clear decrease in width, compared to WT (Fig. S8d and e).

Yes, the tiny roots suggested that OsGSTF12 may have additional functions. It was unknown that whether OsGSTF12 functions in anthocyanin accumulation or flavonoid transport like GSTF12 of cotton or *A. thaliana*.

Fig. S8 Characterization of basic properties of OsGSTF12-KO transgenic plants. d Grain phenotypes of lines #12 and #29, as well as WT in the background of *O. sativa* L.ssp. *Japonica*, variety *Nipponbare*. Bars, 10 mm. **(e)** Measurement of grain length, width, and weight of the WT, #12 and #29 lines. Ns, not significant.

Line 378-383-To claim that this is first report showing role of exosomes in RDV transmission, Following experiments must be addressed:

- a) Authors need to provide quantification on exosome numbers from non-infected and RDV-infected leafhoppers to claim that exosomes are induced upon rDV infection.
- b) Evidence to show that RDV is inside but not outside exosomes is needed.
- c) Evidence for absence of RDV in saliva is needed to claim that it is only present in exosomes.

Response: This is not the first report showing role of exosomes in RDV transmission. In previous study, we published paper titled as “Exosomes mediate horizontal transmission of viral pathogens from insect vectors to plant phloem” in eLife (2021). We provided quantification on exosome numbers from uninfected and RDV-infected leafhoppers to claim that exosomes are induced upon RDV infection, showed that RDV virions are inside but not outside exosomes in electron microscopy, and used GW4869 or dsRab27 to inhibit exosomes to claim the function of exosome in RDV delivery.

Reference: Chen Q, Liu Y, Ren J, Zhong P, Chen M, Jia D, Chen H, Wei T. Exosomes mediate horizontal transmission of viral pathogens from insect vectors to plant phloem. eLife. 2021, 10: e64603.

d) Quantification of NcVg inside and outside exosomes is needed to claim that RDV-infected leafhoppers secrete most or all of NcVg in exosomes but not in saliva.

Response: NcVg inside exosomes of RDV-infected and uninfected salivary glands was respectively quantified in immunofluorescence microscopy and immunoelectron microscopy, as shown in Fig. 2c and d. In immunofluorescence microscopy, we found that RDV infection in salivary glands increased the number of NcVg antigens colocalizing with NcRab27a. It was suggested that RDV-infected salivary glands likely secrete more NcVg in exosomes than in saliva.

In immunoelectron microscopy, we quantified the vesicles associated with NcVg in cavities. We found that the number of vesicles associated with NcVg antibody in cavity per section of one type III-cell of RDV-infected salivary glands was significantly higher than that of uninfected salivary glands. The related contents were shown in Fig. 2d-ii and on lines 205-209.

Fig. 2c-ii The mean number of NcVg colocalizing with NcRab5 or NcRab27a in uninfected or infected salivary gland. Means (\pm SD) represent 3 biological replicates (two-tailed t-test). V+, infected salivary glands. V-, uninfected salivary glands.

Fig. 2d-ii The mean number of vesicles associated with NcVg antibody in cavities per section of one type III-cell of one uninfected or infected salivary gland are shown. Fifteen random samples from infected or uninfected salivary glands were examined (two-tailed t-test).

Reviewer #3 (Remarks to the Author):

The authors have addressed most of the comments.

However, I noted that there is a discrepancy in the data in Figure 4E from the earlier submission to the revised submission. The authors need to explain the reason for this discrepancy in the data. Actually, the data presented in both versions for the same data is opposite.

Reviewer #3 (Remarks to the Author):

The authors have addressed most of the comments.

However, I noted that there is a discrepancy in the data in Figure 4E from the earlier submission to the revised submission. The authors need to explain the reason for this discrepancy in the data. Actually, the data presented in both versions for the same data is opposite.

Response: We are terribly sorry for this revision. We found that the data of V- and V+ in Figure 4E of the earlier submission was exchanged by mistakes, therefore, we corrected this mistake in the revised submission.